# EFFICIENT MULTI-OBJECTIVE PROMPT OPTIMIZATION VIA PURE-EXPLORATION BANDITS

**Donghao Li**[*]  **Chengshuai Shi**[†]  **Weijuan Ou**[◇]  **Cong Shen**[*]  **Jing Yang**[*]

[*] University of Virginia, Charlottesville, VA 22904, USA

[†] Princeton University, Princeton, NJ 08544, USA

[◇] Southern University of Science and Technology, Shenzhen, Guangdong 518055, China

## ABSTRACT

Prompt engineering has become central to eliciting the capabilities of large language models (LLMs). At its core lies *prompt selection* – efficiently identifying the most effective prompts. However, most prior investigations overlook a key challenge: the inherently multi-faceted nature of prompt performance, which cannot be captured by a single metric. To fill this gap, we study the multi-objective prompt selection problem under two practical settings: Pareto prompt set recovery and best feasible prompt identification. Casting the problem into the pure-exploration bandits framework, we adapt provably efficient algorithms from multi-objective bandits and further introduce a novel design for best feasible arm identification in structured bandits, with theoretical guarantees on the identification error in the linear case. Extensive experiments across multiple LLMs show that the bandit-based approaches yield significant improvements over baselines, establishing a principled and efficient framework for multi-objective prompt optimization.

## 1 INTRODUCTION

Prompt engineering has become a practical way to leverage large language models (LLMs) without expensive, time-consuming fine-tuning (Sahoo et al., 2024; Schulhoff et al., 2024). Early results show that zero-shot (Radford et al., 2019) and few-shot prompting (Brown et al., 2020) with a small number of examples can elicit strong performance from frozen models. More recently, chain-of-thought (CoT) prompting (Wei et al., 2022; Kojima et al., 2022; Zhang et al., 2023) has further unlocked step-by-step reasoning, matching or even surpassing task-specific fine-tuning on several complex benchmarks. Prompting has also become foundational across adjacent areas, including retrieval-augmented generation (RAG) (Kang et al., 2024), text-to-image generation (Brade et al., 2023; Mo et al., 2024), and jailbreak analysis and defenses (Yan et al., 2024; Mehrotra et al., 2024; Xu & Parhi, 2025).

Generally speaking, prompt engineering comprises *manual prompt design*, in which experts craft prompts through intuition and iteration, and *automatic prompt optimization*, in which algorithms search the prompt space systematically. Examples of the latter include evolutionary search for high-performing prompts (Guo et al., 2024); gradient-based methods such as AutoPrompt and APO (Shin et al., 2020; Pryzant et al., 2023); reinforcement learning approaches that cast prompt construction as sequential decision making (Deng et al., 2022); and methods that use an LLM itself as the optimizer (Cheng et al., 2024; Tang et al., 2025). Despite their algorithmic differences, these methods share a core challenge: prompt selection, i.e., *given a finite candidate prompt set $\mathcal{X}$ and a limited evaluation budget $B$, how should one allocate queries to identify the prompts that maximize task performance?*

Prior work on prompt selection spans Bayesian optimization (Chen et al., 2024; Sabbatella et al., 2024), discrete search (Hu et al., 2024), and bandit formulations (Shi et al., 2024; Lin et al., 2024b;a; Kong et al., 2025). Despite these advances, existing methods predominantly optimize a *single* objective (e.g., task accuracy), leaving broader trade-offs among multiple objectives largely addressed.

However, in many real-world applications, prompt performance is inherently multi-faceted, involving *multiple* objectives rather than a single metric. For example, in text summarization tasks, human and benchmark assessments consider coherence, faithfulness, fluency, and relevance, and no single metric captures all dimensions (Fabbri et al., 2021). In text style transfer tasks, outputs must satisfy both content preservation and style adherence (e.g., modern-to-Shakespearean translation (Caldas et al., 2018) and politeness transfer (Madaan et al., 2020)). In these multi-objective settings, a single prompt usually cannot be universally superior across all metrics. The trade-offs among these objectives require moving beyond scalarized prompt selection to procedures that preserve multiple criteria throughout the evaluation.

Meanwhile, research on multi-objective bandits offers principled tools for trade-off exploration under fixed evaluation budgets, including algorithms that learn with multiple criteria and handle explicit metric constraints with instance-dependent guarantees (Auer et al., 2016; Kone et al., 2024; 2025; Faizal & Nair, 2022). Yet this toolkit has not been systematically applied to multi-objective prompt selection.

In this work, we aim to bridge these areas by *formulating multi-objective prompt selection within a bandit framework and developing a principled, bandit-based approach for efficient prompt selection under stringent prompt evaluation budget constraint*. Our major contributions are three-fold:

- First, we bridge prompt selection under multiple evaluation criteria with the framework of multi-objective bandits. To the best of our knowledge, this is the first attempt to formalize multi-criteria prompt selection as a multi-objective bandit problem. This connection allows us to move beyond ad-hoc or single-metric prompt selection strategies and instead leverage the rich toolbox of multi-objective bandit algorithms. By doing so, we obtain a principled and efficient framework for balancing diverse evaluation criteria, leading to more robust and systematic prompt selection.

- Secondly, within this framework, we investigate two fundamental problems: best feasible prompt identification and Pareto prompt set identification. For the former, we introduce a general algorithm, GENSEC, and provide a theoretical characterization of its error rate under the linear reward setting. For the latter, we propose another general algorithm, GENPSI, which unifies and generalizes existing bandit algorithms for both the standard and linear settings. Importantly, both GENSEC and GENPSI are designed to accommodate general shared structures among prompts, thereby enhancing learning efficiency, particularly when the evaluation budget is limited.

- We evaluate the performance of GENSEC and GENPSI on two summarization benchmarks: XSum and CNN/DailyMail. For best feasible prompt identification, GENSEC based algorithms recover over 80% and 90% of the utility of the optimal constrained prompt on the two tasks, respectively, whereas the baseline methods achieve only 20–50%. For Pareto prompt set identification, GENPSI based algorithms recover more than 90% of the hypervolume of the ground-truth Pareto set, while the baselines remain in the low-to-mid 80% range.

## 2 RELATED WORKS

**Single-objective prompt selection.** InstructZero (Chen et al., 2024) applies Bayesian optimization in the continuous space of soft prompts, while Sabbatella et al. (2024) uses Bayesian optimization over hard prompts by modeling prompts as $n$-gram phrases. ZOPO (Hu et al., 2024) specializes in selection from a fixed candidate set and argues that a locally optimal prompt can outperform generation-based algorithms. Bandit-based methods improve sample efficiency via adaptive allocation and elimination (Shi et al., 2024; Lin et al., 2024b;a; Kong et al., 2025): TRIPLE (Shi et al., 2024) casts selection as fixed-budget best-arm identification; INSTINCT (Lin et al., 2024b) leverages NeuralUCB with transformer features to optimize instructions; APOHF (Lin et al., 2024a) frames human-in-the-loop selection as a dueling bandit over pairwise preferences; and EXPO (Kong et al., 2025) addresses non-stationarity in agentic settings with adversarial bandits.

**Multi-objective prompt engineering.** Multi-criteria prompt optimization has been receiving increasing attention due to its practical impact across various domains. Evolutionary approaches, such as EMO-Prompts (Baumann & Kramer, 2024), InstOptima (Yang & Li, 2023), and MOPO (Resendiz & Klinger, 2025) use LLM-driven mutation and crossover to generate candidate prompts with high-quality trade-offs. Beyond evolutionary search, MORL-Prompt (Jafari et al., 2024) adapts multi-objective RL with Pareto-aware policy gradient, and GEPA (Agrawal et al., 2025) combines

natural-language reflection with Pareto-guided evolution. *While those work aim to achieve trade-offs among different metrics in prompt optimization, they mostly focus on prompt generation while adopting uniform sampling as the selection method.*

For constrained prompt optimization, CAPO (Zehle et al., 2025) adds cost awareness by combining evolutionary search with a length penalty, yielding prompts that balance task accuracy against token usage. Co-Prompt (Cho et al., 2023) focuses on token-wise prompt generation optimization, and uses another discriminator model to evaluate the generated token. *To the best of our knowledge, there are currently no prompt optimization studies that consider hard constraints, which is a key focus of our work.*

**Multi-objective bandits.** *Pareto set identification* in multi-armed bandits was first studied in the fixed-confidence setting (Auer et al., 2016). In the fixed-budget regime, Kone et al. (2024) introduced Empirical Gap Elimination (EGE), and showed that the misidentification probability decays exponentially with the budget, achieving the optimal rate up to constants. Kone et al. (2025) further investigated the linear bandits setting, while Zuluaga et al. (2013; 2016) investigate the problem from a Bayesian perspective. In the *constrained bandits* setting, regret minimization was investigated in Kagrecha et al. (2023); Pacchiano et al. (2021), while pure exploration strategies have been studied in Faizal & Nair (2022); Bian & Tan (2025) recently. *In summary, there has been active research on multi-objective bandits, providing a rich set of tools for principled prompt optimization. However, existing work primarily focuses on bandits with simple structures, which limits their applicability in practical scenarios. Our work builds on the bandit framework and extends it with application-inspired designs to address these gaps.*

## 3 MULTI-OBJECTIVE PROMPT SELECTION

We first introduce the basic notations used in the prompt-assisted interactions with LLMs (Zhou et al., 2022; Chen et al., 2024; Shi et al., 2024). Let $x$ be a prompt, $\mathcal{D} = \{(q, a)\}$ a task dataset consisting of inputs $q$ and reference answers $a$, and $\mathcal{M}$ a black-box LLM that maps the prompt $x$ together with an input $q$ to a distribution over the output language space $\mathcal{Y}$. Given $(x, q)$, the LLM generates outputs according to $y \sim \mathcal{M}(q; x)$, where $y \in \mathcal{Y}$.

Due to the multi-criteria nature of prompt performance (Baumann & Kramer, 2024; Yang & Li, 2023; Agrawal et al., 2025), we consider $m$ objectives represented by evaluation functions

$$f_j : \mathcal{Y} \times \mathcal{Y} \to \mathbb{R}, \quad j = 1, \ldots, m, \tag{1}$$

where each $f_j$ assigns a numerical score to an LLM output $y \in \mathcal{Y}$ given a reference answer $a \in \mathcal{Y}$. For each prompt $x \in \mathcal{X}$, its expected performance vector is

$$\mu(x) = \mathbb{E}_{(q,a)\sim\mathcal{D}} \, \mathbb{E}_{y\sim\mathcal{M}(q;x)} \big(f_1(y, a), \ldots, f_m(y, a)\big) \in \mathbb{R}^m. \tag{2}$$

Here, the inner expectation averages over the stochasticity of the LLM given a fixed input, while the outer expectation averages over the dataset. Thus $\mu(x)$ summarizes how prompt $x$ performs across all objectives on average. In addition, we assume that all metrics are defined such that larger values indicate better performance.

**Motivating example.** Figure 1 illustrates the evaluation of prompts on the XSum dataset using the Llama3 model, where each point corresponds to a prompt's ROUGE score (x-axis) and brevity score (y-axis). These two metrics exhibit inherent trade-offs, motivating joint consideration for prompt optimization.

In this work, we focus on the problem of multi-criteria prompt selection: given a set of candidate prompts $\mathcal{X}$ and evaluation metrics $f_j$, the objective is to identify a subset of prompts from $\mathcal{X}$ that achieves the desired trade-offs. We consider this problem under the following two settings: (i) best feasible prompt identification , and (ii) Pareto prompt set identification.

**Best feasible prompt identification.** The first target arises when one objective is designated as the *primary* performance

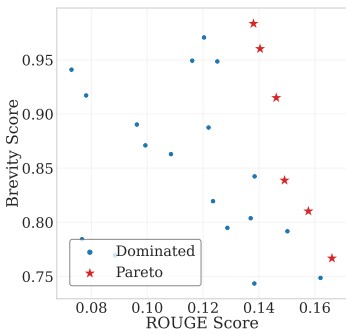

Figure 1: Trade-offs between ROUGE and Brevity.

metric (e.g., task accuracy), while the remaining objectives act
as constraints (e.g., safety above a threshold). Formally, let objective $j = 1$ be the primary objective to maximize, and denote the constraints on the other objectives as $\{\tau_j\}_{j \neq 1}$. A prompt $x$ is *feasible* if

$$\mu_j(x) \geq \tau_j, \quad \forall j \neq 1. \tag{3}$$

The best feasible prompt is then defined as

$$x^\star = \arg\max_{x \in \mathcal{X}} \ \mu_1(x) \quad \text{s.t.} \quad \mu_j(x) \geq \tau_j, \forall j \neq 1. \tag{4}$$

This formulation is especially relevant for safe or regulated domains where utility must be achieved under explicit constraints. In the following designs, we assume that there must exist at least one prompt that is feasible for the considered $\{\tau_j\}_{j \neq 1}$.

**Pareto prompt set identification.** The second target is to recover the Pareto prompt set, which is the set of prompts that is not strictly dominated by any other prompt, as defined in the following.

**Definition 1** (Pareto Optimality). *A prompt $x \in \mathcal{X}$ is* Pareto optimal *if there does not exist another prompt $x' \in \mathcal{X}$ such that $\mu_j(x') \geq \mu_j(x)$ for all $j \in \{1, \ldots, m\}$ and $\mu_j(x') > \mu_j(x)$ for at least one $j$.*

The set of Pareto-optimal prompts is denoted $\mathcal{X}^\star$, and $\{\mu(x) : x \in \mathcal{X}^\star\}$ is the *Pareto front*. In the absence of explicit constraints, Pareto set identification is the most general target for multi-objective optimization, which captures the best achievable trade-offs among objectives.

**Fixed budget constraint.** In practice, prompt evaluations are costly because each trial requires querying an LLM on multiple data examples and metrics. Therefore, in this work, we consider a fixed budget setting, where the total number of evaluations in the optimization procedure is upper bounded by $B$. Then, given a fixed budget constraint $B$, the learner's goal is to identify (i) the optimal feasible prompt $x^*$, or (ii) the Pareto set of prompts $\mathcal{X}^*$, as accurate as possible.

## 4 PROMPT OPTIMIZATION VIA PURE-EXPLORATION BANDITS

Recent work (Shi et al., 2024) shows that prompt optimization can be formulated as a best arm identification (BAI) problem and solved efficiently by leveraging the rich toolbox from BAI in multi-armed bandits. While extensive experiments demonstrate the remarkable performance improvement of such an approach over baselines for a single performance metric, to the best of our knowledge, leveraging bandits algorithms to solve the more general multi-objective prompt optimization has not been studied previously. To fill this gap, in the following, we formulate the multi-objective prompt optimization problem under a fixed budget constraint as a pure-exploration multi-objective bandits.

Specifically, we model the prompt set $\mathcal{X}$ as the set of arms in a multi-armed bandits, and pulling an arm corresponds to evaluating a prompt $x \in \mathcal{X}$ on a randomly sampled input $(q, a) \in \mathcal{D}$, which produces an output $y \sim \mathcal{M}(q; x)$. The evaluation metrics $(f_1(y, a), f_2(y, a), \ldots, f_m(y, a))$ then serve as the stochastic *reward vector* observed for that pull. Thus, the expected reward of each arm is denoted as $\mu(x)$, which captures the mean performance of prompt $x$ across all objectives.

Under the given budget constraint $B$, our objective is to leverage the algorithms from pure-exploration bandits to efficiently identify the optimal feasible prompt and the Pareto set of prompts (arms), respectively.

In practice, the candidate prompt set could be very large. Treating each prompt independently may not be very cost efficient, especially when the total budget is limited. On the hand other, the candidate prompts may inherently have certain correlations, and exploiting such correlation can potentially speed up the learning process and improve the learning accuracy. Motivated by those observations, we introduce a general bandits model to capture such inherent dependency among prompts.

Specifically, we let $\phi : \mathcal{X} \to \mathbb{R}^d$ be a known feature map that embeds a given prompt $x$ to a $d$-dimensional feature vector $\phi(x)$. The expected reward of $x$, denoted as $\mu(x)$, is then equal to $g_\theta(\phi(x))$, where $g_\theta : \mathbb{R}^d \to \mathbb{R}^m$ is a function parameterized by an *unknown* parameter $\theta$. Since $g_\theta$ is

shared across all prompts, the observations obtained by evaluating any of the prompts will contribute to the estimation of $\theta$, potentially speeding up the learning process.

For ease of exposition, in the following, we assume $m = 2$, i.e., there are two metrics for the prompt performance, denoted as $\mu_1$ and $\mu_2$, respectively. The algorithm design and analysis can be extended for a general $m$.

## 5 BEST FEASIBLE PROMPT IDENTIFICATION

For best feasible prompt identification, the learner seeks a prompt that maximizes a primary objective while satisfying feasibility conditions on a secondary criteria, under a fixed budget constraint. Under the corresponding bandits formulation, bandits algorithm Constrained Successive Rejects (CSR) (Faizal & Nair, 2022) is shown to be able to obtain the optimal feasible arm with an exponentially decaying error probability for stochastic bandits.

### 5.1 GENERAL FRAMEWORK

Under the general bandits formulation, we propose a round based arm elimination framework. The process consists of $R$ rounds, each having $n_r$ pulls. In total, it has $\sum_{r=1}^{R} n_r = B$. Each round begins with an active arm set $A_{r-1}$, and at the end of each round, only $l_r$ arms will be kept. $(R, \{n_r\}_{r=1}^{R}, \{l_r\}_{r=1}^{R})$ are determined beforehand through a SCHEDULER, such as Successive Rejection (Audibert & Bubeck, 2010) or Sequential Halving (Karnin et al., 2013). We set $l_R = 1$, which corresponds to the estimated best feasible prompt. As the process proceeds, the major tasks in each round $r$ are as follows.

*(i) Budget allocation.* each round $r$ starts from the active set $A_{r-1}$ and a planned budget $n_r$. The step determines which arms to be pulled for the given budget $n_t$. Denote $(x^{(1)}, \ldots, x^{(n_r)})$ as the list of arms to pull. The budget allocation for the active arms can be flexible as long as a sufficient exploration on the active arms is performed (e.g., uniformly sampling each arm, or adopting G-optimal design to cover the feature space).

*(ii) Arm pulling and evaluation.* For each $t \in [n_r]$, sample $(q_t, a_t) \sim \mathcal{D}$, query the LLM to generate $y_t \sim \mathcal{M}(q_t; x^{(t)})$, evaluate $y_t$ and obtain $f^{(t)} = (f_1(y_t, a_t), f_2(y_t, a_t))$. The new observations $\phi(x^{(t)}, f^{(t)})$ will be included in the collected datasets $X_r$ and $Y_r$.

*(iii) Reward estimation.* An estimator will then utilize $(X_r, Y_r)$ to estimate the unknown parameter $\theta$, based on which the estimated performance $\widehat{\mu}(x)$ can be obtained for each $x \in A_{r-1}$. The estimation can also be performed in flexible ways. In the simplest approach, sample means can be used without accounting for prompt features. More efficient alternatives can be developed based on the considered parameterization function $g_\theta$, e.g., (regularized) least squares.

*(iv) Arm elimination.* Due to the multi-objective setting, the arm elimination step should jointly consider the optimality and feasibility of the active arms. For that purpose, it first forms the empirical feasible arm set $\widehat{\mathcal{F}}_r$ by checking whether $\widehat{\mu}_{2,r}(x) > \tau$. A key component is the elimination step. At the end of round $r$, we first rank the arms by placing the empirical feasible arms before the empirical infeasible arms. Within the empirical feasible set, arms are ordered by decreasing primary reward estimate $\widehat{\mu}_{1,r}(x)$; within the empirical infeasible set, arms are ordered by decreasing constraint estimate $\widehat{\mu}_{2,r}(x)$. We then keep the top $l_r$ arms in this ranking and eliminate the rest, forming the active set $A_r$ for the next round.

Based on the ordering elimination, the learner then eliminates the arms that are determined to be infeasible or suboptimal, and keeps the $l_r$ arms that is more likely to be optimal feasible to form $A_r$. It then proceeds to the next round. The algorithm is presented in Algorithm 1.

### 5.2 THEORETICAL ANALYSIS WITH LINEAR REWARD FUNCTIONS

We instantiate the general framework to a special case where the reward function $g_\theta(\phi(x))$ is linear in both $\phi(x)$ and $\theta$, i.e., $\mu(x) = \phi(x)^\top \theta$, where $\phi(x) \in \mathbb{R}^d$ and $\theta \in \mathbb{R}^{d \times m}$. This recovers the classical linear bandits setting. The following theoretical guarantee can be obtained.

---

**Algorithm 1** GENeralized Successive Elimination under Constraints (GENSEC)

---

1: **Input:** budget $B$, constraint threshold $\tau$, prompt set $\mathcal{X}$, feature map $\phi(\cdot)$, dataset $\mathcal{D}$
2: **Initialization:** $A_0 \leftarrow \mathcal{X}$; $(R, \{n_r\}_{r=1}^R, \{l_r\}_{r=1}^R) \leftarrow \text{SCHEDULER}(K, B)$; $X_0 \leftarrow \emptyset, Y_0 \leftarrow \emptyset$
3: **for** $r = 1 : R$ **do**
4:     $(x^{(1)}, \ldots, x^{(n_r)}) \leftarrow \text{ALLOCATOR}(n_r, A_{r-1})$
5:     At each step $t \in \{1, \ldots, n_r\}$, pull arm $x^{(t)}$ with feature $\phi(x^{(t)})$, collect evaluation $f^{(t)}$, and update observations as $X_r \leftarrow X_{r-1} \cup \{\phi(x^{(t)})\}, Y_r \leftarrow Y_{r-1} \cup \{f^{(t)}\}$
6:     Obtain estimator $\widehat{\mu}_r(x) \leftarrow \text{ESTIMATOR}(x; X_r, Y_r), \forall x \in A_{r-1}$
7:     Construct the empirically feasible set and the empirically infeasible set:

$$\widehat{\mathcal{F}}_r \leftarrow \{x \in A_{r-1} : \widehat{\mu}_{2,r}(x) > \tau\},$$
$$\widehat{\mathcal{F}}_r^c \leftarrow \{x \in A_{r-1} : \widehat{\mu}_{2,r}(x) \leq \tau\}$$

8:     Sort the arms in $\widehat{\mathcal{F}}_r$ in decreasing order of $\widehat{\mu}_{1,r}(x)$, followed by the arms in $\widehat{\mathcal{F}}_r^c$, ordered in decreasing $\widehat{\mu}_{2,r}(x)$
9:     Let $A_r$ consist the first $l_r$ arms in this ordering
10: **end for**
11: **Output:** $A_R$

---

**Theorem 1** (Informal version of Theorem 2). *Assume total budget* $B \geq 45d \lceil \log_2 K \rceil$. *With* SCHEDULER *being Sequential Halving,* ALLOCATOR *using the G-optimal design (Section A.3.1), and* ESTIMATOR *based on least squares, the probability that Algorithm 1 fails to return* $x^\star$ *satisfies* $\Pr\left[x^\star \notin A_R\right] \leq 48 \lceil \log_2 K \rceil \cdot \exp\left\{-\frac{c_1}{dH} \cdot \left\lfloor \frac{B}{\lceil \log_2 K \rceil} \right\rfloor\right\}$, *where* $c_1$ *is a positive constant,* $H = \max_{x \in \mathcal{X} \setminus \{x^*\}} \frac{1}{\Delta(x)^2}$, *and* $\Delta(x)$ *is the constrained gap defined as* $\Delta(x) = \min\{\max(\tau - \mu_2(x), \mu_1(x^*) - \mu_1(x)), \mu_2(x^*) - \tau\}$.

Compared to the existing upper bounds in the stochastic bandit problem (Faizal & Nair, 2022), our result makes a significant improvement in the dependency on $K$. Specifically, while the existing results exhibit a dependence of $K^3$ in the leading coefficient, our upper bound only depends on $\log_2 K$, greatly reducing the impact of $K$ on the error probability. Furthermore, although we adopt a different definition for $H$, both results show similar influence of the budget $B$ and the constrained gap on the error upper bound.

**Proof sketch of Theorem 1.** At a high level, the proof shows that the algorithm rarely eliminates the optimal feasible arm $x^\star$. It consists of the following major steps. **Step 1:** establish uniform concentration bounds for the empirical estimates, which is derived from a self-normalized concentration inequality for linear models as shown in Lemma 1. **Step 2:** use the uniform concentration bounds to show that any suboptimal or infeasible arm appears better than $x^\star$ only with exponentially small probability (see Lemma 2). During this process, **three types of arms** are carefully considered: feasible but suboptimal arms, deceiver arms (infeasible but better than the optimal feasible arm on the primary objective), and infeasible sub-optimal arms, which brings more complicated failure modes compared with previous single-objective analyses. **Step 3:** Lemma 3 argues that $x^\star$ can only be eliminated if many such arms are simultaneously misleading, which is very unlikely. Finally, **Step 4** combines the error probabilities across all rounds via a union bound. This yields an error bound that decays exponentially in the budget $B$, up to logarithmic factors.

## 6   PARETO PROMPT SET IDENTIFICATION

In this section, we investigate the Pareto set identification problem and propose an algorithm named Generalized Pareto Set Identification (GENPSI), which can be found in Section B. Compared with Algorithm 1, GENPSI shares the same components such as *budget allocation*, *arm pulling and evaluation*, and *reward estimation*. The major difference lies in the last component, i.e., *arm elimination*. Due to the different objectives between best feasible arm identification and Pareto set identification, we use the empirical Pareto gap (Kone et al., 2024) as the metric for arm elimination.

We denote the empirical Pareto set as $\widehat{\mathcal{X}}^\star$, and employ the following notations

$$\widehat{m}(x,y) = \min_{i \in [m]} \widehat{\mu}_i(y) - \widehat{\mu}_i(x), \qquad \widehat{M}(x,y) = \max_{i \in [m]} \widehat{\mu}_i(x) - \widehat{\mu}_i(y),$$

$$\widehat{\delta}^+(x) = \min_{y \in \widehat{\mathcal{X}}^\star \setminus \{x\}} \min\left(M(x,y), M(y,x)\right),$$

$$\widehat{\delta}^-(x) = \min_{y \notin \widehat{\mathcal{X}}^\star} \left( \max(\widehat{M}(y,x), 0) + \max_{y' \in \widehat{\mathcal{X}}^\star} \widehat{m}(y,y') \right).$$

Then, the empirical Pareto gap is defined as

$$\widehat{\Delta}(x) = \begin{cases} \max_{y \in \widehat{\mathcal{X}}^\star} \widehat{m}(x,y), & x \notin \widehat{\mathcal{X}}^\star, \\ \min\{\widehat{\delta}^+(x), \widehat{\delta}^-(x)\}, & x \in \widehat{\mathcal{X}}^\star. \end{cases}$$

The Pareto gap measures the difficulty of classifying an arm to be Pareto or sub-optimal.

We note that GENPSI recovers the Empirical Gap Elimination (EGE) (Kone et al., 2024) for stochastic bandits and G-optimal Empirical Gap Elimination (GEGE) (Kone et al., 2025) for linear bandits when the corresponding ALLOCATOR and ESTIMATOR are set in the same form.

## 7 EXPERIMENTS

**Datasets.** We evaluate the prompt selection algorithms on two standard summarization benchmarks: **XSum** (Narayan et al., 2018) and **CNN/DailyMail** (Hermann et al., 2015). The XSum dataset contains approximately 227,000 documents paired with concise one-sentence summaries, whereas the CNN/DailyMail dataset comprises roughly 311,000 article-summary pairs with multi-sentence outputs.

**Candidate prompt generation.** We generate candidate prompts $\mathcal{X}$ using LLaMA-3. For each dataset, 200 prompts are generated based on randomly sampled examples from the dataset (Zhou et al., 2022). The generated prompts are then manually filtered to remove some completely irrelevant prompts and down-sampled to form the final prompt pool of size 100. Within the prompt pool, we further sample 50 and 30 prompts to form smaller candidate prompt sets.

**Models.** We evaluate the performance of candidate prompts on two LLMs: instruction-tuned LLaMA-3-8B-instruct and Gemma-7B-it. Given a prompt $x$ and a query $q$, the LLM generates an output using greedy decoding with a maximum of 512 new tokens, which will then be evaluated according the metrics below.

**Metrics.** We evaluate LLM generated responses on two metrics: **ROUGE-L F1** (Lin, 2004), measuring the overlap with the reference summary, and **Brevity score**, measuring the token lengths. The details of the reward definition can be found in Section C.2.

**Feature extraction.** To facilitate the general bandits formulation, we use GPT-3.5-turbo to extract the embeddings of the prompts. Let $e(x) \in \mathbb{R}^p$ be the extracted embedding for prompt $x \in \mathcal{X}$. We then perform principle component analysis (PCA) to obtain a reduced feature representation. Let $U_d \in \mathbb{R}^{p \times d}$ denote the matrix of the top-$d$ eigenvectors of the sample covariance of $\{e(x)\}_{x \in \mathcal{X}}$. The reduced feature of $x$ is then $\phi(x) = U_d^\top \left(e(x) - \bar{e}\right) \in \mathbb{R}^d$, where $\bar{e} = \frac{1}{K} \sum_{x \in \mathcal{X}} e(x)$.

### 7.1 BEST FEASIBLE PROMPT IDENTIFICATION

In this subsection, we evaluate our constrained prompt selection algorithm, GENSEC, where the objective is to maximize task utility under a prescribed brevity constraint. We instantiate GENSEC into two variants, namely, CSR and MLP-CSR.

**CSR.** We first treat prompts independently, and adopt Successive Rejection, uniform pulling and sample averaging as the Scheduler, Allocator and Estimator, respectively, under which GENSEC reduces to CSR.

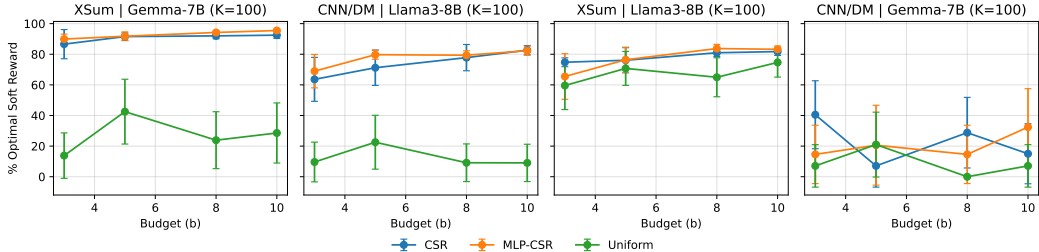

Figure 2: Average soft constrained reward vs. per-arm budget. Error bars denote 95% confidence intervals over 20 random seeds.

Table 1: Average soft constrained reward on XSum - Gemma.

| K | Method | b = 3 | b = 5 | b = 8 | b = 10 |
|---|--------|-------|-------|-------|--------|
| 30 | Uniform | $0.015 \pm 0.010$ | $0.000 \pm 0.000$ | $0.030 \pm 0.013$ | $0.037 \pm 0.014$ |
| | CSR | $0.117 \pm 0.013$ | $0.137 \pm 0.007$ | $\mathbf{0.144 \pm 0.000}$ | $\mathbf{0.143 \pm 0.000}$ |
| | MLP-CSR | $\mathbf{0.123 \pm 0.010}$ | $\mathbf{0.139 \pm 0.002}$ | $0.140 \pm 0.002$ | $0.142 \pm 0.001$ |
| 50 | Uniform | $0.021 \pm 0.011$ | $0.021 \pm 0.011$ | $0.037 \pm 0.014$ | $0.036 \pm 0.014$ |
| | CSR | $0.122 \pm 0.012$ | $\mathbf{0.143 \pm 0.001}$ | $0.141 \pm 0.002$ | $0.140 \pm 0.002$ |
| | MLP-CSR | $\mathbf{0.143 \pm 0.002}$ | $0.142 \pm 0.002$ | $\mathbf{0.147 \pm 0.001}$ | $\mathbf{0.142 \pm 0.002}$ |
| 100 | Uniform | $0.021 \pm 0.011$ | $0.066 \pm 0.016$ | $0.037 \pm 0.014$ | $0.044 \pm 0.015$ |
| | CSR | $0.134 \pm 0.007$ | $0.141 \pm 0.002$ | $0.142 \pm 0.001$ | $0.143 \pm 0.002$ |
| | MLP-CSR | $\mathbf{0.139 \pm 0.002}$ | $\mathbf{0.142 \pm 0.002}$ | $\mathbf{0.145 \pm 0.001}$ | $\mathbf{0.147 \pm 0.001}$ |

**MLP-CSR.** For the general bandits setting, we use an MLP consisting of a ReLU neural network with one hidden layer of 30 hidden states to model the reward function $g_\theta(\cdot)$. We set the Scheduler to be Sequential Halving to reduce the training rounds.

Denote the final output of the algorithms as $\hat{x}$. Then, we define *soft constrained reward* of the $\hat{x}$ as $\mu_1(\hat{x})$ if $\mu_2(\hat{x}) \geq 0.9\tau$; otherwise, it equals zero. We use this definition to tolerate slight violation of the constraint. In Figure 2, we report the average soft constrained reward as a function of the average budget per arm with $K = 100$ prompts. We normalize the value by $\mu_1(x^*)$, where $x^*$ is the best feasible arm. We note that the proposed bandits based algorithms outperform the uniform baseline in almost all settings. The advantage is more significant when performing task CNN_Dailymail on Llama3 (denoted as 'CNN_Dailymail - Llama3') and performing task XSUM on Gemma (denoted as 'XSum - Gemma'). For both cases, when the average budget on each arm $b$ is sufficiently large, the proposed bandits based algorithms recover more than 80% and 90% of the utility of $x^*$ subject to the relaxed constraint, while the baseline only reaches 20% to 50% of that, respectively.

In Table 1, we present the soft constrained reward results by performing task XSum on Gemma. Across different prompt set sizes $K$, and per-arm budget $b$, both CSR and MLP-CSR consistently outperform the uniform pulling baseline. The uniform pulling baseline rarely finds a feasible near-optimal prompt, while our algorithms can reliably find a close-optimal prompt. Notably, MLP-CSR yields slightly higher average soft rewards than CSR, demonstrating the advantage of shared structure in reward functions.

## 7.2 PARETO PROMPT SET IDENTIFICATION

The proposed GENPSI framework for Pareto prompt set identification is also instantiated into two variants.

**EGE.** We first treat prompts independently, and adopt Successive Rejection, uniform pulling and sample averaging as the Scheduler, Allocator and Estimator, respectively, under which GENPSI reduces to EGE.

**MLP-EGE.** Following the same configuration of Scheduler and reward function as in MLP-CSR, we instantiate GENPSI to MLP-EGE.

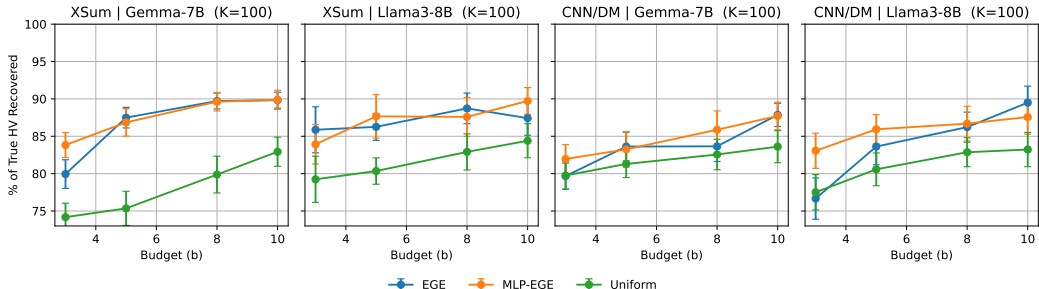

Figure 3: Hypervolume recovery vs. per-arm budget. Error bars denote 95% confidence intervals over 20 random seeds.

Table 2: Hypervolume (HV) on CNN/DailyMail - Llamma 3.

| K | Method | b = 3 | b = 5 | b = 8 | b = 10 |
|---|--------|-------|-------|-------|--------|
| 30 | Uniform | $0.1577 \pm 0.0039$ | $0.1534 \pm 0.0051$ | $0.1685 \pm 0.0028$ | $0.1661 \pm 0.0037$ |
| | EGE | $0.1603 \pm 0.0049$ | $0.1680 \pm 0.0028$ | $0.1753 \pm 0.0033$ | $\mathbf{0.1744 \pm 0.0043}$ |
| | MLP-EGE | $\mathbf{0.1632 \pm 0.0034}$ | $\mathbf{0.1688 \pm 0.0039}$ | $\mathbf{0.1803 \pm 0.0022}$ | $0.1727 \pm 0.0030$ |
| 50 | Uniform | $0.1559 \pm 0.0029$ | $0.1543 \pm 0.0031$ | $0.1618 \pm 0.0023$ | $0.1646 \pm 0.0031$ |
| | EGE | $\mathbf{0.1651 \pm 0.0023}$ | $0.1647 \pm 0.0022$ | $\mathbf{0.1706 \pm 0.0032}$ | $\mathbf{0.1720 \pm 0.0023}$ |
| | MLP-EGE | $0.1618 \pm 0.0033$ | $\mathbf{0.1628 \pm 0.0027}$ | $0.1671 \pm 0.0028$ | $0.1673 \pm 0.0026$ |
| 100 | Uniform | $0.1519 \pm 0.0024$ | $0.1579 \pm 0.0022$ | $0.1624 \pm 0.0019$ | $0.1631 \pm 0.0023$ |
| | EGE | $0.1503 \pm 0.0028$ | $0.1639 \pm 0.0024$ | $0.1690 \pm 0.0020$ | $\mathbf{0.1754 \pm 0.0022}$ |
| | MLP-EGE | $\mathbf{0.1628 \pm 0.0024}$ | $\mathbf{0.1684 \pm 0.0020}$ | $\mathbf{0.1699 \pm 0.0023}$ | $0.1716 \pm 0.0023$ |

To evaluate the performance of the algorithms on Pareto prompt set identification, we introduce a metric called hypervolume (HV) (Knowles et al., 2003). HV measures the Lebesgue volume of objective space dominated by the obtained Pareto set, which is a scaler indicator of the quality and diversity of trade-offs of the estimated Pareto set. HV is computed with respect to a reference point, which we set to the origin for both metrics. We also present the recovered HV as a percentage of the ground-truth Pareto-set HV, which is obtained by exhaustively evaluating all prompts in the candidate pool.

In Figure 3, we report the results when prompt set size $K = 100$. We note that both bandits based algorithms consistently outperform the baseline. When per-arm budget $b$ is low, MLP-EGE achieves the highest performance on average, indicating the advantage of exploiting shared parameters under GENPSI. When per-arm budget $b = 8, 10$, both elimination approaches recover about 90% HV of the ground-truth Pareto set, whereas the baseline recovers around the low-to-mid 80% range.

Table 2 presents the average hypervolume on XSum dataset with Gemma for varying $K$ and $b$. Bandits based algorithms consistently outperform the baseline, indicating robustness of the proposed GENPSI framework. More comprehensive evaluation results can be found in Section C.4.

# 8    CONCLUSION

In this work, we established a principled connection between multi-objective prompt selection and the framework of multi-objective pure-exploration bandits, representing (to the best of our knowledge) the first formalization of this problem in such a setting. Within this framework, we addressed two fundamental problems: best feasible prompt identification and Pareto prompt set identification. To this end, we proposed two general algorithms, GENSEC and GENPSI, which were theoretically grounded and designed to exploit shared structures among prompts to improve sample efficiency under limited evaluation budgets. Extensive experiments on XSum and CNN/DailyMail demonstrated the effectiveness of our approach. Our framework opened new opportunities for principled and scalable prompt optimization under complex evaluation criteria. Future directions include extending our

methods to more diverse tasks and models, incorporating richer evaluation metrics such as fairness and efficiency, and exploring strategies for real-world, large-scale deployment.

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

APPENDIX

# A  BEST FEASIBLE PROMPT IDENTIFICATION

## A.1  SETTINGS AND NOTATIONS

In this section, we formally present the Linear Constrained Sequential Halving in Algorithm 2. We first recap the linear constrained bandit setting and notations. For the multi-objective bandit setting, we have the prompt or arm set $\mathcal{X}$ with $|\mathcal{X}| = K$, and the expected performance or reward vector $\mu(x), x \in \mathcal{X}$. For each arm, there is an associated feature $\phi(x) \in \mathbb{R}^d$. In the linear constrained bandit setting, we assume the number of reward dimensions is only two for simplicity and the expected rewards satisfy the linear structure,

$$\mu_1(x) = \phi(x)^\top \theta^{(1)}, \qquad \mu_2(x) = \phi(x)^\top \theta^{(2)}, \tag{5}$$

where $\theta^{(1)}, \theta^{(2)} \in \mathbb{R}^d$ are unknown. Moreover, for the theoretical results, we further assume that the one-time evaluation results satisfy,

$$f^{(t)} = (f_1^{(t)}, f_2^{(t)}) = (\mu_1(x^{(t)}), \mu_2(x^{(t)})) + (\eta_1^{(t)}, \eta_2^{(t)}), \tag{6}$$

where $f^{(t)}$ is the evaluation reward of one pull of the arm $x^{(t)}$ and $\eta_1^{(t)}, \eta_2^{(t)}$ are the independent $\sigma$-sub-Gaussian noise.

**Definition 2** ($\sigma$-sub-Gaussian). *A random variable $X$ is* 1-sub-Gaussian *if for all $\lambda \in \mathbb{R}$,*

$$\mathbb{E}\left[e^{\lambda\,(X-\mathbb{E}[X])}\right] \le \exp\left(\frac{\sigma^2 \lambda^2}{2}\right).$$

## A.2  CONSTRAINED GAP

Given a constrained threshold $\tau$, the feasible set is denoted as $\mathcal{F} := \{\, x \in \mathcal{X} : \mu_2(x) \ge \tau \,\}$. The goal of the constrained bandit algorithm is to identify the optimal arm $x^\star$,

$$x^\star := \arg\max_{x \in \mathcal{F}} \mu_1(x).$$

With the notations, we follow Faizal & Nair (2022) to define the violation gap and sub-optimal gap as follows

$$\mathrm{viol}(x) := \max\{\tau - \mu_2(x),\, 0\}, \qquad \mathrm{subopt}(x) := \mu_1(x^\star) - \mu_1(x).$$

They characterize the difficulty to distinguish a prompt as infeasible or suboptimal. Then, the gap used to classify an arm as infeasible or suboptimal is denoted as

$$\delta(x) = \max\{\mathrm{viol}(x),\, \mathrm{subopt}(x)\}.$$

We note that for $x \ne x^*$, $\delta(x) > 0$ while $\delta(x^*) = 0$.

Finally, note that the probability of failing to identify the optimal arm also includes the case where the optimal arm is estimated as infeasible. Therefore, we define the constrained gap as

$$\Delta(x) = \min\{\delta(x),\, \mu_2(x^*) - \tau\}. \tag{7}$$

We also assume that the arms are indexed in ascending order of their constrained gaps, with the first arm corresponding to the optimal feasible arm, such that $\Delta(1) \le \Delta(2) \le \cdots \le \Delta(K)$.

## A.3  ALGORITHM DESIGN

We present the Linear Constrained Sequential Halving Algorithm for the linear reward setting in Algorithm 2. The algorithm is instantiated from GENSEC Algorithm 1 by using the Sequential Halving (SH) scheduler, the G-optimal design allocator and the linear estimator.

---

**Algorithm 2** LINEAR CONSTRAINED SEQUENTIAL HALVING

---

1: **Input:** budget $B$, threshold $\tau$, prompt set $\mathcal{X}$, feature map $\phi(\cdot)$, tolerance $\epsilon$, parameter $\kappa \in (0, 1/3]$
2: **Initialization:** $A_0 \leftarrow \mathcal{X}$; $X_0 \leftarrow \emptyset, Y_0 \leftarrow \emptyset$
3: $(R, \{n_r\}_{r=1}^R, \{l_r\}_{r=1}^R) \leftarrow$ SCHEDULER$(K, B)$:

$$\text{Sequential Halving}: \quad R = \lceil \log_2 K \rceil, \quad l_r = \left\lceil \frac{K}{2^r} \right\rceil, \quad n_r = \left\lfloor \frac{B}{R} \right\rfloor$$

4: **for** $r = 1$ **to** $R$ **do**
5:     Allocate $n_r$ pulls across active arm set $A_{r-1}$ based on G-optimal design (Algorithm 3):

$$(x^{(1)}, \dots, x^{(n_r)}) \leftarrow \text{G-OPTIMAL DESIGN}(n_r, A_{r-1}, \phi, \epsilon, \kappa)$$

6:     Pull the arms and collect the observations:

$$X_r \leftarrow \{\phi(x^{(t)})\}_t, \quad Y_r \in \mathbb{R}^{n_r} \leftarrow \{f^{(t)}\}_t$$

7:     Estimate $\mu$ based on $X_r, Y_r$:

$$\widehat{\mu}_r(x) \leftarrow \text{LINEAR ESTIMATOR}(x; \phi(\cdot), X_r, Y_r), \forall x \in A_{r-1}$$

8:     Construct the empirically feasible set and the empirically infeasible set:

$$\widehat{\mathcal{F}}_r \leftarrow \{x \in A_{r-1} : \widehat{\mu}_{2,r}(x) > \tau\},$$
$$\widehat{\mathcal{F}}_r^c \leftarrow \{x \in A_{r-1} : \widehat{\mu}_{2,r}(x) \le \tau\}$$

9:     Sort the arms in $\widehat{\mathcal{F}}_r$ in decreasing order of $\widehat{\mu}_{1,r}(x)$, followed by the arms in $\widehat{\mathcal{F}}_r^c$, ordered in decreasing $\widehat{\mu}_{2,r}(x)$
10:     Let $A_r$ consist the first $l_r$ arms in this ordering
11: **end for**
12: **Output:** $A_R$

---

### A.3.1 G-OPTIMAL DESIGN

The G-optimal design follows Kone et al. (2025), achieving the same estimation error bound as in Lemma 1 for multi-objective bandits. It employs entropic mirror descent to solve the continuous relaxation of the G-optimal design problem, followed by an efficient rounding step to produce an approximate integer solution. These procedures are adapted from Tao et al. (2018) and Allen-Zhu et al. (2017), and are denoted by MD and ROUND, respectively.

The output $N_i$ of the rounding procedure indicates the number of pulls allocated for arm $i$. The corresponding budget allocation $\{x^{(t)}\}_{t=1}^{n_r}$ can then be constructed accordingly. We present the subroutine below.

---

**Algorithm 3** G-OPTIMAL DESIGN (KONE ET AL., 2025)

---

1: **Input:** budget $N$, active set $A$, feature map $\phi(\cdot)$, tolerance $\epsilon$, parameter $\kappa \in (0, 1/3]$
2: $\{w_i\}_{i \in A} \leftarrow$ MD$(A, \phi, \epsilon)$
3: $\{N_i\}_{i \in A} \leftarrow$ ROUND$(N, \{w_i\}_{i \in A}, \kappa, A, \phi)$
4: **for** $t = 1$ **to** $N$ **do**
    $x^{(t)} \leftarrow i$, if $t \in \left( \sum_{j=1}^{i-1} N_j, \sum_{j=1}^i N_j \right]$
5: **end for**
6: **Output:** $\{x^{(t)}\}_t$

---

### A.3.2 LINEAR ESTIMATOR

For the linear case, the estimator only relies on the data collected in the current round to estimate $\theta$. In round $r$, given the active arm set $A_{r-1}$, the arm pulled at step $t$ satisfies $x^{(t)} \in A_{r-1}$ and yields the observed outcome $f^{(t)}$. We define $X_r$ as the design matrix obtained by stacking the feature

vectors $\phi(x^{(t)})^\top$ from round $r$, and $Y_r$ as the vector of the corresponding outcomes $f^{(t)}$.

$$X_r := \begin{bmatrix} \phi(x^{(1)})^\top \\ \phi(x^{(2)})^\top \\ \vdots \\ \phi(x^{(n_r)})^\top \end{bmatrix} \in \mathbb{R}^{n_r \times d}, \qquad Y_r := \begin{bmatrix} f^{(1)} \\ f^{(2)} \\ \vdots \\ f^{(n_r)} \end{bmatrix} \in \mathbb{R}^{n_r}.$$

In each round $r$, define the Gram matrix

$$V_r := X_r^\top X_r \in \mathbb{R}^{d \times d}.$$

Denoting $\Phi_r = [\,\phi(x_{r,1}), \ldots, \phi(x_{r,k_r})\,] \in \mathbb{R}^{d \times k_r}$ as a maximal linearly independent subset selected from $\{\phi(x)\}_{x \in A_r}$, we can define the pesudo inverse

$$V_r^\dagger := \Phi_r\big(\Phi_r^\top V_r \Phi_r\big)^{-1}\Phi_r^\top \quad \in \mathbb{R}^{d \times d}. \tag{8}$$

Then, the estimation becomes

$$\widehat{\theta}_r = V_r^\dagger X_r^\top Y_r, \qquad \widehat{\mu}_r(x) = \phi(x)^\top \widehat{\theta}_r. \tag{9}$$

### A.4 Proof of Theorem 2

Before we proceed to prove Theorem 2, we first introduce several key lemmas.

We first bound the estimation error in each round by adapting Lemma 2 of Kone et al. (2025).

**Lemma 1** (Adapted from Lemma 2 in Kone et al. (2025)). *Let $n_r \geq 45d_{r-1}$, where $d_r = dim(span\{\phi(x) : x \in A_r\})$, $\theta \in \mathbb{R}^{d \times m}$. Then, under Algorithm 2, for all $\epsilon > 0$ and $x \in A_{r-1}$,*

$$\mathbb{P}\left(\left\|(\theta - \widehat{\theta})^\top \phi(x)\right\|_\infty \geq \epsilon\right) \leq 4\exp\left(-\frac{an_r\epsilon^2}{d_r}\right),$$

*where $a = \frac{1}{6\sigma^2}$.*

Lemma 1 indicates that, by applying the G-optimal allocator and the linear estimator, the estimated rewards concentrate around the true performance mean.

Next, we leverage Lemma 1 to show Lemma 2, which bounds the probability that a sub-optimal prompt is empirically better than the best feasible arm.

**Lemma 2.** *At one round $r$, consider arm $x$ remaining active in the arm set $A_{r-1}$, define event $G_r(x) = \{$arm $x$ ranked higher than arm 1 at the end of round $r\}$. It can be obtained that*

$$\mathbb{P}(G_r(x)) \leq 12\exp\left(-\frac{an_r\Delta(x)^2}{4d_r}\right).$$

*Proof.* Let $F_r(x)$ denote the event that arm $x$ is empirically feasible at the end of round $r$, i.e., $\widehat{\mu}_{2,r}(x) \geq \tau$, and $F_r^c(x)$ be its complement. It can be first obtained that

$$\mathbb{P}(G_r(x)) = \mathbb{P}(G_r(x) \cap F_r^c(1)) + \mathbb{P}(G_r(x) \cap F_r(1))$$
$$\leq \mathbb{P}(F_r^c(1)) + \mathbb{P}(G_r(x) \cap F_r(1)).$$

In the following, we bound the two terms of $\mathbb{P}(F_r^c(1))$ and $\mathbb{P}(G_r(x) \cap F_r(1))$ respectively.

First, for the term $\mathbb{P}(F_r^c(1))$, with Lemma 1, it holds that

$$\mathbb{P}(F_r^c(1)) \leq 4\exp\left(-\frac{an_r(\mu_2(1) - \tau)^2}{d_r}\right).$$

Then, for the term $\mathbb{P}(G_r(x) \cap F_r(1))$, we do a case-by-case analysis based on the property of arm $x$. It is easy to see that arm $x$ must fall into one of the three cases.

**Case 1.** $\mu_1(x) < \mu_1(1)$ **and** $\mu_2(x) > \tau$**.** In this case, it holds that $G_r(x) \cap F_r(1) \subseteq \{\widehat{\mu}_{1,r}(x) > \widehat{\mu}_{1,r}(1)\}$, which implies that

$$\mathbb{P}(G_r(x) \cap F_r(1)) \leq \mathbb{P}(\widehat{\mu}_{1,r}(x) > \widehat{\mu}_{1,r}(1))$$

$$\leq \mathbb{P}\left(\widehat{\mu}_{1,r}(x) - \mu_1(x) \geq \frac{\mu_1(1) - \mu_1(x)}{2}\right)$$

$$+ \mathbb{P}\left(\mu_1(1) - \widehat{\mu}_{1,r}(1) \geq \frac{\mu_1(1) - \mu_1(x)}{2}\right)$$

$$\leq 8 \exp\left(-\frac{an_r(\mu_1(1) - \mu_1(x))^2}{4d_r}\right),$$

where the last inequality is from Lemma 1.

**Case 2.** $\mu_1(x) > \mu_1(1)$ **and** $\mu_2(x) \leq \tau$**.** In this case, it holds that $G_r(x) \cap F_r(1) \subseteq \{\widehat{\mu}_{2,r}(x) > \tau\}$, which leads to

$$\mathbb{P}(G_r(x) \cap F_r(1)) \leq \mathbb{P}(\widehat{\mu}_{2,r}(x) > \tau) \leq 4 \exp\left(-\frac{an_r(\tau - \mu_2(x))^2}{d_r}\right),$$

where the last inequality is from Lemma 1.

**Case 3.** $\mu_1(x) < \mu_1(1)$ **and** $\mu_2(x) \leq \tau$**.** In this case, it holds that $G_r(x) \cap F_r(1) \subseteq \{\widehat{\mu}_{2,r}(x) > \tau\} \cap \{\widehat{\mu}_{1,r}(x) > \widehat{\mu}_{1,r}(1)\}$, which leads to

$$\begin{aligned}
\mathbb{P}(G_r(x) \cap F_r(1)) &\leq \mathbb{P}(\widehat{\mu}_{2,r}(x) > \tau, \widehat{\mu}_{1,r}(x) > \widehat{\mu}_{1,r}(1)) \\
&\leq \min\left\{\mathbb{P}(\widehat{\mu}_{2,r}(x) > \tau), \mathbb{P}(\widehat{\mu}_{1,r}(x) > \widehat{\mu}_{1,r}(1))\right\} \\
&\leq \min\left\{8 \exp\left(-\frac{an_r(\mu_1(1) - \mu_1(x))^2}{4d_r}\right), 4 \exp\left(-\frac{an_r(\tau - \mu_2(x))^2}{d_r}\right)\right\} \\
&\leq 8 \exp\left(-\frac{an_r(\max\{\mu_1(1) - \mu_1(x), \tau - \mu_2(x)\})^2}{4d_r}\right)
\end{aligned}$$

where the third inequality can be obtained similarly as Case 1 and Case 2 following Lemma 1.

Combining the three cases, regardless of the property of arm $x$, it holds that

$$\mathbb{P}(G_r(x) \cap F_r(1)) \leq 8 \exp\left(-\frac{an_r(\max\{\mu_1(1) - \mu_1(x), \tau - \mu_2(x)\})^2}{4d_r}\right).$$

The proof can then be concluded by adding the terms $\mathbb{P}(F_r^c(1))$ and $\mathbb{P}(G_r(x) \cap F_r(1))$ together as

$$\begin{aligned}
\mathbb{P}(G_r(x)) &\leq \mathbb{P}(F_r^c(1)) + \mathbb{P}(G_r(x) \cap F_r(1)) \\
&\leq 4 \exp\left(-\frac{an_r(\mu_2(1) - \tau)^2}{d_r}\right) + 8 \exp\left(-\frac{an_r(\max\{\mu_1(1) - \mu_1(x), \tau - \mu_2(x)\})^2}{4d_r}\right) \\
&\leq 12 \exp\left(-\frac{an_r\Delta(x)^2}{4d_r}\right),
\end{aligned}$$

where the last step comes from the definition of $\Delta(x)$. $\qquad\square$

**Lemma 3.** *The probability that arm $1$ is eliminated on round $r$ satisfies*

$$\mathbb{P}(1 \notin A_r | 1 \in A_{r-1}) \leq 48 \exp\left(-\frac{an_r \min_{i \neq 1} \Delta(x)^2}{4d_r}\right).$$

*Proof.* Denote $N_r$ as the number of arms ranked higher than arm $1$ at the end of round $r$, i.e.,

$$N_r = \sum_{x \in A_{r-1}, x \neq 1} \mathbb{I}\{G_r(x)\},$$

where we reused the definition of $G_r(x)$ in Lemma 2. For arm $1$ to be eliminated at the end of round $r$, the following needs to happen:

$$N_r \geq l_r,$$

i.e., there are at least $l_r$ arms ranked higher than arm $1$. We can bound the probability of this event as

$$\mathbb{P}(N_r \geq l_r) \leq \frac{\mathbb{E}[N_r]}{l_r}$$

$$= \frac{1}{l_r} \sum_{x \in A_{r-1}, x \neq 1} \mathbb{P}(G_r(x))$$

$$\leq \frac{1}{l_r} \sum_{x \in A_{r-1}, x \neq 1} 12 \exp\left(-\frac{an_r \Delta(x)^2}{4d_r}\right)$$

$$\leq \frac{12 l_{r-1}}{l_r} \cdot \exp\left(-\frac{an_r \min_{x \neq 1} \Delta(x)^2}{4d_r}\right)$$

$$\leq 48 \exp\left(-\frac{an_r \min_{x \neq 1} \Delta(x)^2}{4d_r}\right)$$

where the first inequality follows the Markov inequality and the second inequality leverages Lemma 2. The proof is then concluded. $\qquad\square$

**Theorem 2.** *Given a fixed set of $K$ arms $\mathcal{X}$ with the linear reward structure specified in Equations* (5) *and* (6), *under total budget $B \geq 45d \lceil \log_2 K \rceil$, the probability that Algorithm 2 fails to output the best feasible arm is upper bounded by*

$$\mathbb{P}(1 \notin A_R) \leq 48 \lceil \log_2 K \rceil \exp\left(-\frac{a}{4} \cdot \left\lfloor \frac{B}{\lceil \log_2 K \rceil} \right\rfloor \cdot \frac{1}{dH}\right),$$

*where*

$$a = \frac{1}{6\sigma^2}, \quad H = \max_{x \in \mathcal{X} \setminus \{1\}} \frac{1}{\Delta^2(x)},$$

*and $\Delta(x)$ is defined in Equation* (7).

*Proof.* With Lemma 3, the proof of the theorem can be obtained as follows:

$$\mathbb{P}(1 \notin A_R) = \sum_{r=1}^{R} \mathbb{P}(1 \notin A_r, 1 \in A_{r-1})$$

$$\leq \sum_{r=1}^{R} \mathbb{P}(1 \notin A_r | 1 \in A_{r-1})$$

$$\leq \sum_{r=1}^{R} 48 \exp\left(-\frac{an_r \min_{x \neq 1} \Delta(x)^2}{4d_r}\right)$$

$$\leq 48 \lceil \log_2 K \rceil \exp\left(-\frac{a}{4} \cdot \left\lfloor \frac{B}{\lceil \log_2 K \rceil} \right\rfloor \cdot \frac{1}{dH}\right),$$

which concludes the proof. $\qquad\square$

## B   PARETO SET IDENTIFICATION

### B.1   RESTATE THE GENPSI

For the GENPSI framework, the main difference from GENSEC is the design of gaps. In GENPSI, we use the Pareto gap, following Auer et al. (2016) and Kone et al. (2024). To define the gap, we first introduce the following defintions.

**Definition 3** (Pareto Dominance). *We say that arm $x$ Pareto-dominates arm $y$ (denoted as $\mu(x) \succ \mu(y)$) if and only if*

$$\forall i, \ \mu_i(x) \geq \mu_i(y) \ and \ \exists i, \ \mu_i(x) > \mu_i(y).$$

**Definition 4** (Pareto set). *For a given set of arms $\mathcal{X}$, the Pareto set is defined as*

$$\mathcal{X}^\star := \left\{ x \in \mathcal{X} \ \Big| \ \nexists x' \in \mathcal{X} \ such \ that \ \mu(x') \succ \mu(x) \right\},$$

*i.e., the Pareto set consists of all arms that are not dominated by any other arm.*

---

**Algorithm 4** GENeralized Pareto Set Identification (GENPSI)

---

1: **Input:** budget $B$, prompt set $\mathcal{X}$, feature map $\phi(\cdot)$, dataset $\mathcal{D}$
2: **Initialization:** $A_0 \leftarrow \mathcal{X}$; $(R, \{n_r\}_{r=1}^R, \{l_r\}_{r=1}^R) \leftarrow \text{SCHEDULER}(K, B)$; $X_0 \leftarrow \emptyset, Y_0 \leftarrow \emptyset$
3: **for** $r = 1 : R$ **do**
4: $\quad (x^{(1)}, \ldots, x^{(n_r)}) \leftarrow \text{ALLOCATOR}(n_r, A_{r-1})$
5: $\quad$ At each step $t \in \{1, \ldots, n_r\}$, pull arm $x^{(t)}$ with feature $\phi(x^{(t)}$, observe evaluation $f^{(t)}$, and update observations as $X_r \leftarrow X_{r-1} \cup \{\phi(x^{(t)})\}, Y_r \leftarrow Y_{r-1} \cup \{f^{(t)}\}$
6: $\quad$ Obtain estimator $\widehat{\mu}_r(x) \leftarrow \text{ESTIMATOR}(x; X_r, Y_r), \forall x \in A_{r-1}$
7: $\quad$ Calculate the empirical dominating set

$$\widehat{\mathcal{X}}_r^\star := \left\{ x \in \widehat{\mathcal{X}} \,\middle|\, \nexists x' \in \mathcal{X} \text{ such that such that } \hat{\mu}_r(x') \succ \hat{\mu}_r(x) \right\}$$

8: $\quad$ Estimate the empirical Pareto gap

$$\widehat{\Delta}_r(x) = \begin{cases} \max\limits_{y \in \widehat{\mathcal{X}}^\star} \widehat{m}(x, y), & x \notin \widehat{\mathcal{X}}_r^\star, \\ \min\{\widehat{\delta}^+(x), \widehat{\delta}^-(x)\}, & x \in \widehat{\mathcal{X}}_r^\star \end{cases}$$

$\quad$ where the $\widehat{m}, \widehat{\delta}^+, \widehat{\delta}^-$ are all the empirical version based on $\widehat{\mu}$
9: $\quad$ Perform arm elimination: $A_r \leftarrow \text{ELIMINATOR}\big(A_{r-1}, l_r, \widehat{\Delta}_r(\cdot)\big)$
10: **end for**
11: **Output:** $A_R$

---

Then, we define $m(x, y)$ and $M(x, y)$ for any prompt pair $x, y \in \mathcal{X}$ as follows:

$$m(x, y) := \min_{i \in [m]} \big(\mu_i(y) - \mu_i(x)\big), \qquad M(x, y) := \max_{i \in [m]} \big(\mu_i(x) - \mu_i(y)\big).$$

These two terms quantify the level of $y$ dominating $x$.

If $x \notin \mathcal{X}^\star$, the Pareto gap is directly

$$\Delta(x) = \max_{y \in \mathcal{X}^\star} m(x, y),$$

which quantifies the greatest dominance from a Pareto prompt.

If $x \notin \mathcal{X}^\star$, we further denote

$$\delta^+(x) := \min_{y \in \mathcal{X}^\star \setminus \{x\}} \min\big(M(x, y), M(y, x)\big), \qquad \delta^-(x) := \min_{y \notin \mathcal{X}^\star} \Big(\max\big(M(y, x), 0\big) + \Delta(y)\Big),$$

and define the Pareto gaps for $x \in \mathcal{X}^\star$

$$\Delta(x) = \min\big\{\delta^+(x), \delta^-(x)\big\}.$$

In Algorithm 4, we use the empirical estimates of the Pareto gaps for arm elimination.

## B.2 EXTENSION TO GENERAL $m \geq 2$

Our framework extends naturally to settings with more than two objectives. The key quantities underlying our algorithms, such as the Pareto gaps and constrained gaps, can be generalized to $m$-dimensional objective spaces without changing the fundamental structure of the elimination or selection rules. The computational overhead associated with these extensions is modest: dominance checks and gap computations scale polynomially with $m$ and remain efficient for the small number of objectives typically encountered in practice. Thus, the proposed algorithms retain both conceptual simplicity and computational tractability when applied to $m > 2$ multi-objective prompt selection problems.

## C    EXPERIMENTS DETAILS AND RESULTS

### C.1    MODELS AND TEMPLATE

We uses two target models Gemma-7b-it (Team et al., 2024) and Llama3-8b-instruct (Grattafiori et al., 2024). For the instruction models, we adopt the recommended system template as Figure 4.

```
System instruction template

<|begin_of_text|><|start_header_id|>system<|end_header_id|>

You are an intelligent assistant. Please finish the given task,
answer with the output only and reply nothing else.

<|eot_id|><|start_header_id|>user<|end_header_id|>

<prompt>

<|eot_id|><|start_header_id|>assistant<|end_header_id|>
```

Figure 4: The system prompt template is used for both Gemma and Llama3.

In addition, to generate the prompt sets, we use the generation template as Figure 5.

```
Prompt Generation Template

Input: [Input1]
Output: [Output1]

Input: [Input2]
Output: [Output2]

...

Please provide the instruction now.
```

Figure 5: Prompt generation with 3-5 examples.

### C.2    METRICS USED IN SUMMARIZATION TASKS

In our experiments, we mainly consider two metrics for the summarization task, the ROUGE and Brevity score.

**ROUGE Score.**    The ROUGE (Lin, 2004) score is a widely used metric for the summarization task. It captures the token-wise similarity between two texts. We use its variant, ROUGE-Lsum, provided by the Hugging Face **evaluate** python package. In general, it computes the longest common subsequences between the generated text and the reference, and aggregates the results across sentences. It captures the similarity between the generated summary from the LLM and the gold reference from the dataset.

**Brevity Score.**    The Brevity score function is an artificial score to map the token lengths into the range $[0, 1]$. It is a piecewise function

$$f_B(x) = \begin{cases} 1, & x \leq \tau_{\text{low}}, \\ \dfrac{\tau_{\text{high}} - x}{\tau_{\text{high}} - \tau_{\text{low}}}, & \tau_{\text{low}} < x < \tau_{\text{high}}, \\ 0, & x \geq \tau_{\text{high}} \, . \end{cases}$$

where $x$ is the token length and $\tau_{\text{low}} < \tau_{\text{high}}$.

## C.3 IMPLEMENTATION DETAILS WITH GENERAL REWARD STRUCTURE

We now specify the algorithm for general reward functions. We set the SCHEDULER to be Sequential Halving. A uniform budget ALLOCATOR is adopted in each round for ease of implementation.

For the ESTIMATOR, a neural network with parameter $\theta$ is used to approximate the reward function $g_\theta$. In each round $r$, we use the observations collected from the pulled arms to optimize $\theta$. Specifically, we let

$$\mathcal{L}_r(\theta; \lambda) = \frac{1}{n_r} \sum_{t=1}^{n_r} \|g_\theta(\phi^{(t)}) - f^{(t)}\|_2^2 + \lambda \|\theta\|_2^2. \tag{10}$$

where the $\ell_2$ regularization is adopted to prevent model overfitting.

Then, the estimator obtains the estimates according to

$$\hat{\theta}_r = \arg\min_\theta \mathcal{L}_r(\theta; \lambda), \quad \widehat{\mu}_r(x) = g_{\hat{\theta}_r}(\phi(x)), \quad \forall x \in A_r.$$

## C.4 EXPERIMENTAL RESULTS

### C.4.1 BEST FEASIBLE PROMPT IDENTIFICATION

We list our main results of best feasible prompt identification in Table 3 and Table 4, which are separately for datasets XSum and CNN/DailyMail. As defined in the main paper, $K$ denotes the size of the candidate prompts, and $b$ denotes the budget per arm. We report the averaged soft constrained reward that *soft constrained reward* defined as $\mu_1(\hat{x})$ if $\mu_2(\hat{x}) \geq 0.9\tau$, and zero otherwise.

In Tables 3 and 4, CSR and MLP-CSR consistently outperform uniform evaluation across budgets, candidate-set sizes $K$, models, and datasets, highlighting the benefit of adaptive allocation under constraints. As expected, increasing the per-arm budget $b$ generally improves the performance of all methods. The effect of increasing $K$ is mixed. On one hand, a larger candidate pool makes identification more challenging under a fixed budget. On the other hand, it increases the likelihood of containing higher-quality feasible prompts, so the soft-constrained reward does not exhibit a clear monotonic trend with respect to $K$. Finally, uniform evaluation may fail to return any feasible prompt in more challenging regimes, whereas CSR and MLP-CSR remain robust, consistently recovering feasible prompts with high reward.

### C.4.2 PARETO PROMPT SET IDENTIFICATION

Here, we list our main results for Pareto prompt set identification in Table 5 and Table 6, corresponding to the datasets XSum and CNN/DailyMail, respectively.

From Table 5 and Table 6, EGE and especially MLP-EGE generally achieve higher hypervolumn (HV) than uniform evaluation across budgets $b$, prompt pool size $K$, model, and datasets. Especially, MLP-CSR usually outperforms CSR when the prompt size is large and the budget is limited. As expected, increasing the budget would generally improve all methods, since the estimates would be more accurate. The effect of increasing $K$ is also mixed, since larger $K$ makes identification harder, but it can also contain better trade-offs. Generally, elimination-based methods remain robust across settings.

## C.5 ABLATION STUDIES

### C.5.1 DISTRIBUTION OF PROMPT OBJECTIVES

We visualize the objective distributions of the generated prompts to highlight the trade-off between the two metrics and to validate the reasonableness of our prompt-generation process and experimental design.

Figure 6a shows the objective distribution of the 100 generated prompts used in our main XSum experiments. The trade-off between the two metrics is clearly visible, demonstrating that identify-

Table 3: Average soft constrained reward on **XSum**.

| K | Method | b = 3 | b = 5 | b = 8 | b = 10 |
|---|--------|-------|-------|-------|--------|
| | | *Gemma-7B* | | | |
| 30 | Uniform | $0.015 \pm 0.010$ | $0.000 \pm 0.000$ | $0.030 \pm 0.013$ | $0.037 \pm 0.014$ |
| | CSR | $0.117 \pm 0.013$ | $0.137 \pm 0.007$ | $\mathbf{0.144 \pm 0.000}$ | $\mathbf{0.143 \pm 0.000}$ |
| | MLP-CSR | $\mathbf{0.123 \pm 0.010}$ | $\mathbf{0.139 \pm 0.002}$ | $0.140 \pm 0.002$ | $0.142 \pm 0.001$ |
| 50 | Uniform | $0.021 \pm 0.011$ | $0.021 \pm 0.011$ | $0.037 \pm 0.014$ | $0.036 \pm 0.014$ |
| | CSR | $0.122 \pm 0.012$ | $\mathbf{0.143 \pm 0.001}$ | $0.141 \pm 0.002$ | $0.140 \pm 0.002$ |
| | MLP-CSR | $\mathbf{0.143 \pm 0.002}$ | $0.142 \pm 0.002$ | $\mathbf{0.147 \pm 0.001}$ | $0.142 \pm 0.002$ |
| 100 | Uniform | $0.021 \pm 0.011$ | $0.066 \pm 0.016$ | $0.037 \pm 0.014$ | $0.044 \pm 0.015$ |
| | CSR | $0.134 \pm 0.007$ | $0.141 \pm 0.002$ | $0.142 \pm 0.001$ | $0.143 \pm 0.002$ |
| | MLP-CSR | $\mathbf{0.139 \pm 0.002}$ | $\mathbf{0.142 \pm 0.002}$ | $\mathbf{0.145 \pm 0.001}$ | $\mathbf{0.147 \pm 0.001}$ |
| | | *Llama3-8B* | | | |
| 30 | Uniform | $0.048 \pm 0.016$ | $0.069 \pm 0.019$ | $0.080 \pm 0.020$ | $0.089 \pm 0.020$ |
| | CSR | $\mathbf{0.161 \pm 0.003}$ | $\mathbf{0.154 \pm 0.004}$ | $\mathbf{0.148 \pm 0.002}$ | $\mathbf{0.149 \pm 0.003}$ |
| | MLP-CSR | $0.143 \pm 0.003$ | $0.126 \pm 0.010$ | $0.147 \pm 0.004$ | $\mathbf{0.149 \pm 0.002}$ |
| 50 | Uniform | $0.063 \pm 0.018$ | $0.045 \pm 0.017$ | $0.051 \pm 0.018$ | $0.078 \pm 0.019$ |
| | CSR | $\mathbf{0.144 \pm 0.011}$ | $0.134 \pm 0.010$ | $0.122 \pm 0.014$ | $\mathbf{0.157 \pm 0.002}$ |
| | MLP-CSR | $0.141 \pm 0.008$ | $\mathbf{0.151 \pm 0.003}$ | $\mathbf{0.154 \pm 0.003}$ | $0.156 \pm 0.002$ |
| 100 | Uniform | $0.113 \pm 0.015$ | $0.134 \pm 0.010$ | $0.123 \pm 0.012$ | $0.141 \pm 0.009$ |
| | CSR | $\mathbf{0.141 \pm 0.003}$ | $\mathbf{0.144 \pm 0.008}$ | $0.153 \pm 0.002$ | $0.154 \pm 0.002$ |
| | MLP-CSR | $0.124 \pm 0.001$ | $\mathbf{0.144 \pm 0.008}$ | $\mathbf{0.158 \pm 0.003}$ | $\mathbf{0.157 \pm 0.002}$ |

(a) 100 prompts

(b) 1000 prompts

Figure 6: Distributions of the objectives for generated prompts on Xsum.

ing the Pareto-optimal prompts or the optimal feasible prompt is indeed a meaningful task in this environment.

We further extend the prompt set to 1000 candidates in Figure 6b. For this larger pool, no manual filtering is applied. Comparing the two figures, we observe that manual filtering primarily removes prompts that perform poorly on both Brevity and ROUGE, while preserving the overall structure of the distribution. Moreover, the Pareto-front shapes in both figures are similar, indicating that manual filtering does not materially alter the key characteristics of the candidate prompt set.

Table 4: Average soft constrained reward on **CNN/DailyMail**.

| K | Method | b = 3 | b = 5 | b = 8 | b = 10 |
|---|--------|-------|-------|-------|--------|
| | | *Gemma-7B* | | | |
| 30 | Uniform | $0.021 \pm 0.014$ | $0.032 \pm 0.017$ | $0.021 \pm 0.014$ | $0.021 \pm 0.014$ |
| | CSR | $0.081 \pm 0.018$ | $0.134 \pm 0.020$ | $0.153 \pm 0.012$ | $\mathbf{0.166 \pm 0.010}$ |
| | MLP-CSR | $\mathbf{0.138 \pm 0.016}$ | $\mathbf{0.164 \pm 0.011}$ | $\mathbf{0.164 \pm 0.005}$ | $0.163 \pm 0.005$ |
| 50 | Uniform | $0.007 \pm 0.007$ | $0.020 \pm 0.013$ | $0.020 \pm 0.013$ | $0.000 \pm 0.000$ |
| | CSR | $0.091 \pm 0.019$ | $0.161 \pm 0.013$ | $0.161 \pm 0.010$ | $\mathbf{0.167 \pm 0.011}$ |
| | MLP-CSR | $\mathbf{0.146 \pm 0.015}$ | $0.162 \pm 0.008$ | $\mathbf{0.172 \pm 0.007}$ | $0.153 \pm 0.006$ |
| 100 | Uniform | $0.015 \pm 0.015$ | $\mathbf{0.045 \pm 0.022}$ | $0.000 \pm 0.000$ | $0.015 \pm 0.015$ |
| | CSR | $\mathbf{0.088 \pm 0.023}$ | $0.015 \pm 0.015$ | $\mathbf{0.062 \pm 0.024}$ | $0.032 \pm 0.021$ |
| | MLP-CSR | $0.032 \pm 0.020$ | $\mathbf{0.045 \pm 0.027}$ | $0.032 \pm 0.020$ | $\mathbf{0.072 \pm 0.026}$ |
| | | *Llama3-8B* | | | |
| 30 | Uniform | $0.000 \pm 0.000$ | $0.000 \pm 0.000$ | $0.008 \pm 0.008$ | $0.000 \pm 0.000$ |
| | CSR | $\mathbf{0.157 \pm 0.002}$ | $\mathbf{0.136 \pm 0.015}$ | $\mathbf{0.148 \pm 0.011}$ | $\mathbf{0.154 \pm 0.009}$ |
| | MLP-CSR | $0.142 \pm 0.011$ | $0.126 \pm 0.015$ | $0.142 \pm 0.011$ | $\mathbf{0.154 \pm 0.004}$ |
| 50 | Uniform | $0.015 \pm 0.010$ | $0.016 \pm 0.011$ | $0.008 \pm 0.008$ | $0.010 \pm 0.010$ |
| | CSR | $\mathbf{0.154 \pm 0.000}$ | $\mathbf{0.157 \pm 0.009}$ | $0.155 \pm 0.009$ | $\mathbf{0.165 \pm 0.003}$ |
| | MLP-CSR | $0.144 \pm 0.005$ | $0.155 \pm 0.004$ | $\mathbf{0.160 \pm 0.003}$ | $0.160 \pm 0.004$ |
| 100 | Uniform | $0.019 \pm 0.013$ | $0.044 \pm 0.017$ | $0.018 \pm 0.012$ | $0.018 \pm 0.012$ |
| | CSR | $0.123 \pm 0.014$ | $0.138 \pm 0.011$ | $0.151 \pm 0.008$ | $\mathbf{0.160 \pm 0.003}$ |
| | MLP-CSR | $\mathbf{0.134 \pm 0.011}$ | $\mathbf{0.154 \pm 0.003}$ | $\mathbf{0.154 \pm 0.003}$ | $\mathbf{0.160 \pm 0.003}$ |

### C.5.2 VARYING EVALUATION BUDGET

We next analyze how performance changes as the evaluation budget per prompt-selection run increases.

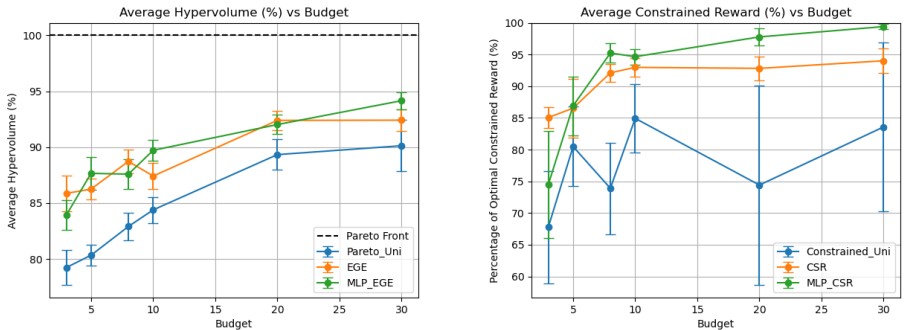

Figure 7: Average recovered hypervolume (%) and constrained reward on XSum using Llama-3.

Figure 7 illustrates how the recovered hypervolume changes as we increase the evaluation budget for both our methods and the baselines. As expected, all algorithms benefit from larger budgets, but our methods consistently outperform the uniform-pulling baseline across all settings. This confirms that increasing the budget does not alter the overall conclusions. In the main experiments, we focus on smaller budgets in order to highlight that our algorithms remain effective even in the low-budget regime, which is the setting of primary interest.

Table 5: Hypervolume (HV) on **XSum**.

| K | Method | b = 3 | b = 5 | b = 8 | b = 10 |
|---|---|---|---|---|---|
| | | *Gemma-7B* | | | |
| 30 | Uniform | $0.1105 \pm 0.0019$ | $0.1127 \pm 0.0021$ | $0.1165 \pm 0.0013$ | $0.1173 \pm 0.0012$ |
| | EGE | $0.1115 \pm 0.0020$ | $\mathbf{0.1172 \pm 0.0014}$ | $\mathbf{0.1188 \pm 0.0015}$ | $0.1193 \pm 0.0013$ |
| | MLP-EGE | $\mathbf{0.1147 \pm 0.0021}$ | $0.1138 \pm 0.0023$ | $0.1172 \pm 0.0018$ | $\mathbf{0.1195 \pm 0.0015}$ |
| 50 | Uniform | $0.1105 \pm 0.0023$ | $0.1169 \pm 0.0019$ | $0.1181 \pm 0.0017$ | $0.1192 \pm 0.0014$ |
| | EGE | $\mathbf{0.1126 \pm 0.0028}$ | $\mathbf{0.1180 \pm 0.0018}$ | $\mathbf{0.1207 \pm 0.0019}$ | $\mathbf{0.1212 \pm 0.0016}$ |
| | MLP-EGE | $0.1117 \pm 0.0024$ | $0.1152 \pm 0.0025$ | $0.1174 \pm 0.0018$ | $0.1187 \pm 0.0012$ |
| 100 | Uniform | $0.1007 \pm 0.0013$ | $0.1023 \pm 0.0016$ | $0.1084 \pm 0.0017$ | $0.1126 \pm 0.0014$ |
| | EGE | $0.1085 \pm 0.0013$ | $\mathbf{0.1187 \pm 0.0010}$ | $\mathbf{0.1218 \pm 0.0007}$ | $0.1219 \pm 0.0007$ |
| | MLP-EGE | $\mathbf{0.1138 \pm 0.0012}$ | $0.1179 \pm 0.0013$ | $0.1216 \pm 0.0009$ | $\mathbf{0.1220 \pm 0.0009}$ |
| | | *Llama3-8B* | | | |
| 30 | Uniform | $0.1433 \pm 0.0026$ | $0.1481 \pm 0.0023$ | $0.1472 \pm 0.0031$ | $0.1493 \pm 0.0031$ |
| | EGE | $\mathbf{0.1614 \pm 0.0008}$ | $\mathbf{0.1496 \pm 0.0022}$ | $0.1535 \pm 0.0031$ | $\mathbf{0.1560 \pm 0.0015}$ |
| | MLP-EGE | $0.1488 \pm 0.0026$ | $\mathbf{0.1496 \pm 0.0031}$ | $\mathbf{0.1577 \pm 0.0019}$ | $0.1548 \pm 0.0017$ |
| 50 | Uniform | $0.1500 \pm 0.0022$ | $0.1557 \pm 0.0024$ | $0.1586 \pm 0.0020$ | $0.1585 \pm 0.0021$ |
| | EGE | $\mathbf{0.1626 \pm 0.0004}$ | $\mathbf{0.1587 \pm 0.0023}$ | $\mathbf{0.1596 \pm 0.0019}$ | $0.1604 \pm 0.0020$ |
| | MLP-EGE | $0.1520 \pm 0.0020$ | $0.1552 \pm 0.0022$ | $0.1595 \pm 0.0013$ | $\mathbf{0.1616 \pm 0.0021}$ |
| 100 | Uniform | $0.1423 \pm 0.0028$ | $0.1443 \pm 0.0016$ | $0.1489 \pm 0.0022$ | $0.1516 \pm 0.0021$ |
| | EGE | $\mathbf{0.1542 \pm 0.0028}$ | $0.1549 \pm 0.0016$ | $\mathbf{0.1594 \pm 0.0019}$ | $0.1570 \pm 0.0021$ |
| | MLP-EGE | $0.1507 \pm 0.0024$ | $\mathbf{0.1574 \pm 0.0027}$ | $0.1573 \pm 0.0024$ | $\mathbf{0.1611 \pm 0.0016}$ |

### C.5.3 EXPERIMENTS WITH AN EXPANDED CANDIDATE PROMPT POOL

We next examine how the algorithms behave when we increase the size of the candidate prompt pool.

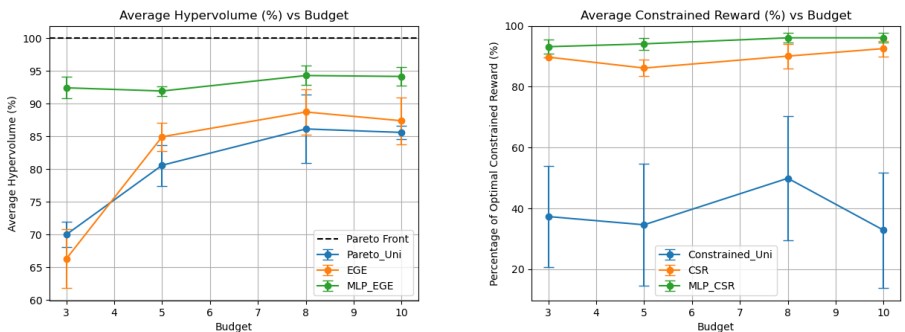

Figure 8: Average recovered hypervolume (%) and average constrained reward (%) with 1000 candidate prompts on XSum using Llama-3.

Figure 8 reports the recovered hypervolume when scaling the candidate pool to 1000 prompts under different evaluation budgets. We observe that MLP-EGE performs best in this larger-pool setting, and EGE outperforms the uniform baseline when the budget is sufficiently large. These results indicate that our algorithms remain robust as the prompt pool grows substantially in size.

Table 6: Hypervolume (HV) on **CNN/DailyMail**.

| K | Method | b = 3 | b = 5 | b = 8 | b = 10 |
|---|--------|-------|-------|-------|--------|
| | | *Gemma-7B* | | | |
| 30 | Uniform | $0.1345 \pm 0.0036$ | $0.1393 \pm 0.0029$ | $0.1484 \pm 0.0019$ | $0.1459 \pm 0.0021$ |
| | EGE | $0.1327 \pm 0.0038$ | $0.1458 \pm 0.0026$ | $0.1472 \pm 0.0023$ | $0.1453 \pm 0.0036$ |
| | MLP-EGE | $\mathbf{0.1441 \pm 0.0029}$ | $\mathbf{0.1499 \pm 0.0019}$ | $\mathbf{0.1505 \pm 0.0016}$ | $\mathbf{0.1505 \pm 0.0019}$ |
| 50 | Uniform | $0.1310 \pm 0.0038$ | $0.1390 \pm 0.0023$ | $0.1414 \pm 0.0021$ | $0.1429 \pm 0.0020$ |
| | EGE | $0.1334 \pm 0.0032$ | $0.1413 \pm 0.0020$ | $\mathbf{0.1455 \pm 0.0024}$ | $0.1445 \pm 0.0019$ |
| | MLP-EGE | $\mathbf{0.1358 \pm 0.0032}$ | $\mathbf{0.1430 \pm 0.0017}$ | $0.1434 \pm 0.0020$ | $\mathbf{0.1463 \pm 0.0016}$ |
| 100 | Uniform | $0.1280 \pm 0.0015$ | $0.1305 \pm 0.0015$ | $0.1325 \pm 0.0017$ | $0.1342 \pm 0.0018$ |
| | EGE | $0.1279 \pm 0.0014$ | $0.1342 \pm 0.0016$ | $0.1343 \pm 0.0016$ | $\mathbf{0.1411 \pm 0.0013}$ |
| | MLP-EGE | $\mathbf{0.1316 \pm 0.0016}$ | $0.1336 \pm 0.0019$ | $\mathbf{0.1379 \pm 0.0021}$ | $0.1408 \pm 0.0015$ |
| | | *Llama3-8B* | | | |
| 30 | Uniform | $0.1577 \pm 0.0039$ | $0.1534 \pm 0.0051$ | $0.1685 \pm 0.0028$ | $0.1661 \pm 0.0037$ |
| | EGE | $0.1603 \pm 0.0049$ | $0.1680 \pm 0.0028$ | $0.1753 \pm 0.0033$ | $\mathbf{0.1744 \pm 0.0043}$ |
| | MLP-EGE | $\mathbf{0.1632 \pm 0.0034}$ | $\mathbf{0.1688 \pm 0.0039}$ | $\mathbf{0.1803 \pm 0.0022}$ | $0.1727 \pm 0.0030$ |
| 50 | Uniform | $0.1559 \pm 0.0029$ | $0.1543 \pm 0.0031$ | $0.1618 \pm 0.0023$ | $0.1646 \pm 0.0031$ |
| | EGE | $\mathbf{0.1651 \pm 0.0023}$ | $0.1647 \pm 0.0022$ | $\mathbf{0.1706 \pm 0.0032}$ | $\mathbf{0.1720 \pm 0.0023}$ |
| | MLP-EGE | $0.1618 \pm 0.0033$ | $\mathbf{0.1628 \pm 0.0027}$ | $0.1671 \pm 0.0028$ | $0.1673 \pm 0.0026$ |
| 100 | Uniform | $0.1519 \pm 0.0024$ | $0.1579 \pm 0.0022$ | $0.1624 \pm 0.0019$ | $0.1631 \pm 0.0023$ |
| | EGE | $0.1503 \pm 0.0028$ | $0.1639 \pm 0.0024$ | $0.1690 \pm 0.0020$ | $\mathbf{0.1754 \pm 0.0022}$ |
| | MLP-EGE | $\mathbf{0.1628 \pm 0.0024}$ | $\mathbf{0.1684 \pm 0.0020}$ | $\mathbf{0.1699 \pm 0.0023}$ | $0.1716 \pm 0.0023$ |

### C.5.4 VARYING CONSTRAINT

Finally, we vary the constraint threshold and track the feasible average reward under different constraints in Figure 9. It indicates that our proposed algorithms consistently outperform the uniform baseline, and the choice of constraint does not compromise the effectiveness of our algorithms.

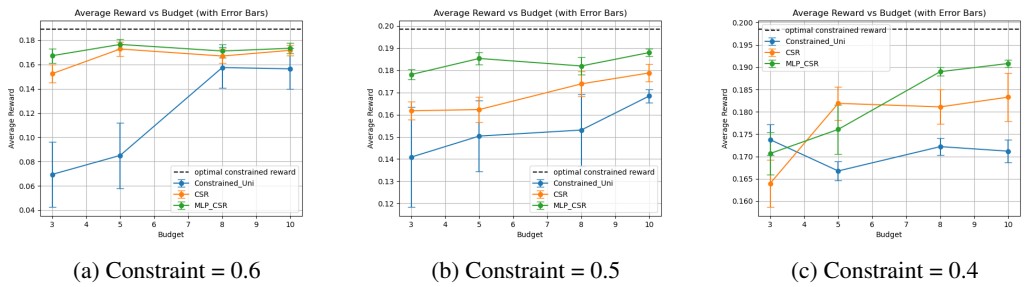

(a) Constraint = 0.6      (b) Constraint = 0.5      (c) Constraint = 0.4

Figure 9: Feasible average reward vs. per-armed budget on XSum using Llama3 with varying constraints.

### C.6 COMPUTING RESOURCES AND COSTS

Our experiments are conducted on a server equipped with an AMD EPYC 9554 CPU, 755 GiB of system memory, and four NVIDIA H100 PCIe GPUs (80 GiB HBM each; driver 550.144.03) running CUDA 12.4.131. We employ Gemma and Llama-3 in quantized, inference-only settings. Generally, the experiments are expected to be reproduced on a server with over 20G GPU memory.

## D  LANGUAGE MODEL USAGE IN PAPER WRITING DISCLOSURE

ChatGPT 5 is used during the writing of the paper, mainly for paraphrasing the sentences, improving the grammar, reorganizing the sentences in paragraphs, and generating a few paragraphs based on human-provided outlines. All the generated texts are reviewed and revised by humans. All technical claims, definitions, algorithms, theorems, and proofs are written by humans.

