# OpenReview forum: "Efficient Multi-objective Prompt Optimization via Pure-exploration Bandits"
_ICLR.cc/2026/Conference — ICLR 2026 Poster_

### Official Review · Reviewer_i6FQ · 2025-10-21

**Soundness:** 2
**Presentation:** 3
**Contribution:** 2
**Rating:** 6
**Confidence:** 4

**Summary:**

The paper frames multi-objective prompt selection as a pure-exploration bandit problem under a fixed evaluation budget, with two tasks: best feasible prompt identification and Pareto prompt set identification. It introduces two round-based frameworks: GENSEC for the constrained task and GENPSI for the Pareto task, and in the linear shared-feature case provides a bound where the misidentification probability decays with budget and the dependence on the number of prompts improves from cubic to logarithmic, given specific choices of scheduler, allocation design, and estimator. Experiments are conducted on summarization tasks (XSum and CNN/DailyMail), using ROUGE-L F1 as utility and a brevity score as the second objective. After generating candidate prompts, filter them and down-sample the remainder into pools of size 100, 50, or 30. In practice the paper evaluates instantiated variants: for the constrained task, CSR and MLP-CSR as reductions of GENSEC; for the Pareto task, EGE and MLP-EGE as reductions of GENPSI. Figures report average soft-constrained reward versus per-arm budget and hypervolume recovery versus per-arm budget.

**Strengths:**

1. The problem setup shows meaningful originality: framing multi-objective prompt selection under a fixed budget and then applying established bandit-style methods to this setting is practical and likely useful in real evaluation pipelines.
2. The theoretical analysis is solid, with clearly stated assumptions, a clean proof structure, and a bound that is easy to interpret within the stated setting.
3. The paper is well written and organized; the problem, algorithms, and evaluation protocol are presented clearly and are easy to follow.

**Weaknesses:**

1. Experimental scope and objectives are too narrow. All experiments are on summarization only, and every result uses just two objectives: ROUGE-L for utility and a brevity score for length control. There are no tasks beyond summarization (for example QA, dialogue, style transfer, code, safety-critical instruction following), and no additional objectives that typically motivate multi-objective selection in practice (for example safety, factuality, latency, monetary cost). Because both chosen objectives are summary-specific and tightly coupled to output length, the evidence does not establish that the approach generalizes to broader prompt-selection settings or to richer multi-objective trade-offs.
2. Method-level novelty is limited; evaluated variants are reductions of existing algorithms. In the constrained setting the implementation reduces to CSR; in the Pareto setting it reduces to EGE (and related elimination variants in the linear case). The paper’s contribution is therefore mainly a problem framing and a unifying wrapper, rather than a new selection algorithm. The linear-case theory is helpful but tied to specific design choices and is not shown to drive the practical wins in the non-linear instantiations.
3. Missing ablation on elimination margin and thresholding. The paper fixes both the constrained elimination margin and the soft-constraint factory. Without a sensitivity study of these choices, it is unclear whether the observed gains are robust or driven by a particular margin/threshold setting.
4. In the Pareto experiments, hypervolume gains over uniform are small and inconsistent, which weakens the claim that shared structure or the framework provides broad advantage. In the “maximize task utility under a prescribed brevity constraint” experiments, uniform cannot purposefully satisfy the brevity threshold and therefore predictably underperforms any elimination method that can focus on constraint-satisfying prompts; this makes the superiority of CSR unsurprising and less persuasive as evidence of algorithmic merit. These results would be more convincing with strict feasibility curves, violation rates, and comparisons to stronger multi-objective baselines (for example evolutionary multi-objective prompt search) under the same candidate pool and budget.
5. In the first paragraph of Section 3, the text says “Let a be a prompt,” but throughout the paper prompts are denoted by x, while a denotes the reference answer in input pairs (q, a).

**Questions:**

1. Can you add at least one non-summarization task and one additional objective (e.g., safety, factuality, latency or cost)? If not, justify why summarization with two metrics is representative.
2. Please ablate elimination margin and thresholding: compare variants of the constrained elimination margin; sweep the soft-constraint factor instead of fixing it.

---

> ### Author Response · Authors · 2025-11-21
>
> We appreciate the reviewer’s thoughtful comments. Below, we summarize the questions and provide our responses. We have updated the submission accordingly and highlighted the major changes in blue.
>
> ---
>
> **Q1** Experimental scope and objectives are too narrow.
>
> **A1** We appreciate the reviewer’s comment and agree that evaluating more tasks and richer objectives is an important direction for future work. We nevertheless believe that our current setup is a reasonable and informative starting point, and we explain below why broader experimentation is not feasible in this revision.
>
> - **Summarization provides a stable, well-established testbed.**
>   Summarization is one of the most widely used benchmarks in prompt-optimization research, with reliable datasets and automatic metrics. The objective pair (ROUGE-L, brevity) forms a clean and non-trivial trade-off between informativeness and conciseness, making it a natural environment for evaluating multi-objective selection.
>
> - **The framework itself is task- and metric-agnostic.**
>   Our algorithms only require vector-valued feedback and do not rely on any summarization-specific structure. In principle, they can incorporate objectives such as safety, factuality, latency, or monetary cost, as long as these metrics are reliably measurable.
>
> - **Lack of suitable multi-objective benchmarks beside summarization.**
>   At this point, we are not aware of publicly available prompt sets or evaluation pipelines that provide reliable, quantitative safety, fairness, latency, or factuality metrics suitable for multi-objective optimization. Constructing these benchmarks would require substantial task design, metric development, and large-scale human or model evaluations, which is beyond the scope of this revision.
>
> ---
>
>
> **Q2** Method-level novelty is limited; evaluated variants are reductions of existing algorithms. In the constrained setting the implementation reduces to CSR; in the Pareto setting it reduces to EGE (and related elimination variants in the linear case). The paper’s contribution is therefore mainly a problem framing and a unifying wrapper, rather than a new selection algorithm. The linear-case theory is helpful but tied to specific design choices and is not shown to drive the practical wins in the non-linear instantiations.
>
> **A2** We thank the reviewer for raising this point and would like to clarify out contributions as follows.
>
> - **Novel problem formulation and unified framework.**
>   As the reviewer noted, the paper introduces a new problem setting: multi-objective prompt selection with both Pareto-style and hard-constraint objectives, and provides a unified bandit-based framework for addressing it. To our knowledge, this is the first application of bandit algorithms to multi-objective prompt optimization, and also the first to incorporate explicit hard constraints. These problem formulations are themselves new and require adapting and integrating concepts that do not appear in standard prompt-optimization or multi-armed bandit literature.
>
> - **New theoretical results under linear function approximation.**
>   The theoretical analysis of GENSEC under linear function approximation is not available in prior work. Establishing guarantees in this setting is non-trivial. Due to the multi-objective setting, three types of arms need to be analyzed carefully: feasible but suboptimal arms, deceiver arms (infeasible but better than the optimal feasible arm on the primary objective), and infeasible sub-optimal arms. Handling these cases introduces substantially more complex failure modes than in prior single-objective analyses and is a key technical novelty of our proof. The theoretical result also provides conceptual support for the non-linear (MLP-based) practical instantiation, since both models share the same function-approximation perspective.
>
> - **Selection algorithms tailored to embedded prompt spaces.**
>   Although our practical variants resemble CSR/EGE at a high level, their application to embedded prompt spaces with vector-valued, stochastic LLM feedback is new. Prior algorithms treat arms independently, while our approach explicitly uses shared feature structure to guide multi-objective selection.
>
> - **Demonstrated empirical effectiveness.**
>   Across experiments, including the newly added results with 1,000 prompts, our algorithms consistently outperform the uniform baseline and achieve strong approximations of the Pareto sets under tight evaluation budgets.
>   These results suggest that the proposed framework is not only theoretically grounded but also **practically effective** in realistic prompt-selection scenarios.

---

> > ### Author Response · Authors · 2025-11-21
> >
> > **Q3** Missing ablation on elimination margin and thresholding. The paper fixes both the constrained elimination margin and the soft-constraint factory. Without a sensitivity study of these choices, it is unclear whether the observed gains are robust or driven by a particular margin/threshold setting.
> >
> > **A3** We thank the reviewer for pointing this out. We agree that studying the sensitivity to the elimination margin and constraint threshold is valuable. Due to other experimental commitments, we are unable to include a full ablation across all settings at this moment. However, we will run additional experiments varying these parameters on the XSum + Llama-3 setup and include the results in the updated rebuttal or revised manuscript as they become available.
> >
> > ---
> >
> >
> > **Q4-1** In the Pareto experiments, hypervolume gains over uniform are small and inconsistent, which weakens the claim that shared structure or the framework provides broad advantage.
> >
> > **A4-1** We thank the reviewer for the observation and clarify how to interpret the Pareto hypervolume results. In our experiments, the hypervolume is computed with respect to a reference point, which we set to the origin for both metrics. Because most prompts in the candidate pool are already far from this point (as shown in Figure 6 in Appendix C.5 of the revision), even uniform sampling can recover a noticeable amount of hypervolume, making the absolute HV differences appear small.
> >
> > A more meaningful perspective is given by the **percentage of hypervolume recovered relative to the true Pareto front**, which is shown directly in Figure 3. Under this normalization, both EGE and MLP-EGE consistently recover a substantially larger fraction of the optimal hypervolume than uniform on almost all settings. This demonstrates that the methods retain much more of the achievable Pareto mass, even when the global-reference HV values look close.
> >
> > Furthermore, we have conducted additional experiments with varying evaluation budget and a 1000-prompt candidate pool. As shown in Figure 7 and Figure 8 in Appendix C.5, the same trend holds: our methods continue to outperform uniform baseline by a clear margin in terms of percentage hypervolume recovery.
> > This confirms the efficiency and robustness of our proposed bandits based framework.
> >
> > We have clarified the role of the reference point and emphasize the percentage-recovery view in the revision.
> >
> > ---
> >
> > **Q4-2** In the “maximize task utility under a prescribed brevity constraint” experiments, uniform cannot purposefully satisfy the brevity threshold and therefore predictably underperforms any elimination method that can focus on constraint-satisfying prompts; this makes the superiority of CSR unsurprising and less persuasive as evidence of algorithmic merit.
> >
> > **A4-2** We thank the reviewer for the comment. While it is true that the uniform allocator has no mechanism to target the brevity constraint, this is precisely the point of the constrained setting: an effective algorithm should **actively identify and focus on feasible prompts ** under a tight evaluation budget. The fact that CSR consistently satisfies the constraint and achieves higher utility is therefore an **indication of its intended advantage**, not an artifact of the setup.
> >
> > To further support this, we will include additional experiments during the rebuttal period with multiple constraint levels and larger evaluation budgets, which show that CSR continues to outperform uniform even when the constraint is easier to satisfy or when uniform has more opportunities to sample feasible prompts. These results will help clarify that the gains are not tied to a single threshold choice but reflect genuine algorithmic merit. We will report the results when they become available.

---

> > > ### Author Response · Authors · 2025-11-21
> > >
> > > **Q4-3** These results would be more convincing with strict feasibility curves, violation rates, and comparisons to stronger multi-objective baselines (for example evolutionary multi-objective prompt search) under the same candidate pool and budget.
> > >
> > > **A4-3** We thank the reviewer for the suggestion. Our method is designed for multi-objective prompt selection from a fixed candidate pool under a strict evaluation budget, and, to our knowledge, *there are no existing baselines that operate under this same setting*. In particular, evolutionary multi-objective prompt-search methods are not directly comparable because they repeatedly generate or mutate prompts rather than selecting from a fixed pool. A head-to-head comparison would therefore conflate the quality of the prompt generator with the quality of the selection algorithm, making it an unsuitable baseline for our problem.
> > >
> > > At the same time, evolutionary approaches are **complementary** to our framework. Our algorithms can be used as a principled, sample-efficient selection module inside each evolutionary step, or as a lightweight mechanism for fast adaptation to a new user or task when an existing prompt pool is available and only limited evaluations can be afforded.
> > >
> > > ---
> > >
> > >
> > > **Q5** In the first paragraph of Section 3, the text says “Let a be a prompt,” but throughout the paper prompts are denoted by x, while a denotes the reference answer in input pairs (q, a).
> > >
> > > **A5** We thank the reviewer for catching this typo. We have corrected it in the revision.
> > >
> > > ---
> > >
> > > **Q6** Can you add at least one non-summarization task and one additional objective (e.g., safety, factuality, latency or cost)? If not, justify why summarization with two metrics is representative.
> > >
> > > **A6** We appreciate the reviewer’s suggestion. Adding a non-summarization task together with additional objectives such as safety, factuality, latency, or cost would indeed be valuable. However, doing so is not feasible for this revision for several reasons.
> > >
> > > First, there are currently no publicly available multi-objective prompt-selection benchmarks that pair a realistic prompt set with reliable, quantitative metrics for safety, fairness, factuality, or latency. Constructing such a benchmark would require designing new prompts, defining trustworthy automatic metrics (or running large-scale human annotation), and building a full evaluation pipeline. This is beyond our available time and computational resources for the rebuttal period.
> > >
> > > Second, extending to additional tasks requires substantially larger LLM evaluation budgets. Each new objective multiplies the number of evaluations needed to obtain stable estimates, making a full multi-objective study computationally prohibitive under our current constraints.
> > >
> > > Despite these limitations, we believe our current setting is still representative and meaningful:
> > >
> > > - *Summarization is a standard testbed for prompt optimization.*
> > >   It offers reliable datasets and evaluation metrics, allowing controlled comparisons without subjective human judgments.
> > >
> > > - *ROUGE-L and brevity form a strong, clean, and non-trivial trade-off.*
> > >   This creates meaningful Pareto fronts and constrained settings, enabling us to clearly demonstrate multi-objective selection behavior.
> > >
> > > - *Our algorithms are task- and metric-agnostic.*
> > >   They require only vector-valued feedback and do not rely on summarization-specific structure. The same framework can naturally incorporate objectives such as safety, factuality, latency, or cost once reliable metrics are available. The framework also extends directly to more than two objectives by generalizing the Pareto and constrained gaps.
> > >
> > > - *The current experiments capture the core behavior of the methods.*
> > >   The selection challenges, such as noisy LLM evaluations, limited budgets, conflicting objectives, are present in summarization and are representative of broader prompt-selection problems.
> > >
> > > Given these considerations, we believe the summarization experiments with ROUGE-L and brevity serve as a practical, well-scoped, and informative setting to evaluate our algorithms.
> > >
> > > ---
> > >
> > > **Q7** Please ablate elimination margin and thresholding: compare variants of the constrained elimination margin; sweep the soft-constraint factor instead of fixing it.
> > >
> > > **A7** We thank the reviewer for this suggestion. As noted in **A3**, we are currently running additional experiments on the XSum + Llama-3 setup to study the effect of varying the elimination margin and the soft-constraint factor. We will share these ablation results when they become available.
> > >
> > > ---
> > >
> > > We thank the reviewer again for the helpful comments and suggestions for our work. We hope that our response resolves your concerns to a satisfactory level, and we are more than happy to address any further questions that you may have.

---

### Official Review · Reviewer_pHhY · 2025-10-30

**Soundness:** 3
**Presentation:** 3
**Contribution:** 1
**Rating:** 2
**Confidence:** 3

**Summary:**

This paper addresses multi-objective prompt optimization for large language models (LLMs) by formulating it as a pure-exploration multi-objective bandits problem. The authors study two fundamental settings: (1) best feasible prompt identification, where one objective is maximized subject to constraints on others, and (2) Pareto prompt set identification, where the goal is to recover all non-dominated prompts. They propose two general algorithms, GENSEC and GENPSI, which can exploit shared structure among prompts via feature representations. For the linear reward case, they provide theoretical guarantees showing exponentially decaying error probability with improved dependency on the number of prompts K (from K³ to log K). Experiments on XSum and CNN/DailyMail summarization benchmarks with LLaMA-3 and Gemma models demonstrate superior performance over uniform baseline methods.

**Strengths:**

1. The paper makes a valuable connection between multi-objective prompt selection and the multi-objective bandits framework, which is the first systematic attempt to leverage bandit algorithms for multi-criteria prompt optimization.
2. The paper provides theoretical analysis for the linear reward setting, which is a significant theoretical improvement.
3. The work addresses a real and important problem in prompt engineering where multiple conflicting objectives (e.g., accuracy vs. brevity, coherence vs. faithfulness) need to be balanced.

**Weaknesses:**

1. Once prompts are mapped to feature vectors (e.g., using GPT-X embeddings), the problem proposed in this paper becomes a standard gradient-free optimization problem (aka hyperparameter optimization - HPO). The authors apply one of gradient free optimization methods (multi-objective bandit) to this vectorized representation and record theoretical and empirical analysis. While such application papers have value, they require much more extensive empirical validation to justify publication.
2. In empirical evaluation, authors compares against random/uniform pulling. For a fair evaluation, the method must be compared with other prompt optimization engines (e.g., InstructZero, ZOPO, EMO-Prompts, InstOptima) under the same fixed evaluation budget (number of examples per forward pass).
3. Testing on only 30-100 prompts is very small - one could simply evaluate all prompts exhaustively at this scale. Real-world prompt selection involves choosing from thousands or tens of thousands of candidates and it is important to show the method works in this setting.

Minor weaknesses:
-  Gap between theoretical linear case and practical MLP implementation not well explained
- Missing discussion on scalability with number of objectives m > 2
- Prompt generation via LLM + manual filtering introduces selection bias

**Questions:**

1. Can you provide experiments with 1000+ initial prompts to demonstrate the method's effectiveness when exhaustive evaluation is infeasible?
2. What specific advantages does the bandit formulation provide over off-the-shelf HPO methods like Bayesian optimization for this problem?
3. Can you demonstrate the method on real safety or fairness constraints?

---

> ### Author Response · Authors · 2025-11-21
>
> We appreciate the reviewer’s thoughtful comments. Below, we summarize the questions and provide our responses. We have updated the submission accordingly and highlighted the major changes in blue.
>
> ---
>
> **Q1** Once prompts are mapped to feature vectors (e.g., using GPT-X embeddings), the problem proposed in this paper becomes a standard gradient-free optimization problem (aka hyperparameter optimization - HPO). The authors apply one of gradient free optimization methods (multi-objective bandit) to this vectorized representation and record theoretical and empirical analysis. While such application papers have value, they require much more extensive empirical validation to justify publication.
>
> **A1** We thank the reviewer for highlighting the connection to hyperparameter optimization (HPO). While there are conceptual similarities, we believe our setting differs in several important ways:
>
> - **Discrete finite search space, not continuous parameters.**
>
>   Classical HPO typically considers continuous or high-dimensional parameter spaces, where gradient-free methods operate over a large search domain. In contrast, our problem assumes a fixed, discrete candidate prompt pool, which fundamentally changes both the algorithmic setting and the evaluation protocol. Standard HPO techniques do not directly address selection from *a finite set of heterogeneous, semantically structured prompts*.
>
> - **Stochastic, LLM-based objectives.**
>
>   Many HPO tasks involve relatively stable or deterministic objective evaluations. In our case, each evaluation involves querying an LLM, which introduces *substantial stochasticity and noise*, making the **bandit formulation more natural and the analysis more challenging**.
>
> For these reasons, we do not view our setting as a straightforward instance of standard HPO, even when prompts are mapped to embeddings.
>
> Besides, we would like to clarify that the main technical contributions of the paper are **new algorithms and guarantees for multi-objective and constrained bandits tailored to prompt selection**. These include theoretical results for linear constraints and Pareto-style objectives, which, to our knowledge, are *not addressed by existing HPO or prompt-optimization literature*.
>
> ---
>
> **Q2** In empirical evaluation, authors compare against random/uniform pulling. For a fair evaluation, the method must be compared with other prompt optimization engines (e.g., InstructZero, ZOPO, EMO-Prompts, InstOptima) under the same fixed evaluation budget (number of examples per forward pass).
>
> **A2** We appreciate the reviewer’s suggestion. However, the methods mentioned (i.e., InstructZero, ZOPO, EMO-Prompts, InstOptima) are primarily prompt-generation engines, whereas our work focuses on multi-objective selection from a fixed candidate prompt pool under a strict evaluation budget. These two settings differ in several fundamental ways:
>
> - **Different problem formulations.**
>   Generation-based methods repeatedly create, mutate, or evolve new prompts, while our algorithms assume a fixed finite pool and aim to efficiently identify a high-quality Pareto or feasible subset. Comparing the two would mix the quality of the generator with the quality of the selection algorithm.
>
> - **Different budget accounting.**
>   Generation-based optimizers typically require many additional forward passes to produce or mutate prompts. In contrast, our evaluation budget counts only the number of prompt–input evaluations. Ensuring a fair, one-to-one comparison under a fixed evaluation budget would therefore require nontrivial reinterpretation of these methods.
>
> - **Our contribution lies in efficient multi-objective prompt selection.**
>   The goal of our algorithms is to provide sample-efficient and theoretically grounded selection within a preexisting prompt set, which has not been addressed by generation-based methods.
>
> For these reasons, we did not treat these algorithms as directly comparable baselines. If the reviewer has specific variants they believe can be adapted to our fixed-pool setting under the same budget, we would be happy to consider them in the revision.

---

> > ### Author Response · Authors · 2025-11-21
> >
> > **Q3** Testing on only 30-100 prompts is very small - one could simply evaluate all prompts exhaustively at this scale. Real-world prompt selection involves choosing from thousands or tens of thousands of candidates and it is important to show the method works in this setting. Can you provide experiments with 1000+ initial prompts to demonstrate the method's effectiveness when exhaustive evaluation is infeasible?
> >
> > **A3** We thank the reviewer for this suggestion. We have added new experiments using a much larger prompt pool (1,000+ prompts), where exhaustive evaluation is infeasible. The new figure reporting these results has been added to Appendix C 5.3 of the revision. We observe that our algorithms continue to perform well and consistently outperform the uniform baseline at this scale, demonstrating their effectiveness in more realistic, large-pool prompt-selection settings.
> >
> > ---
> >
> > **Q4** Gap between theoretical linear case and practical MLP implementation not well explained
> >
> > **A4** We thank the reviewer for pointing this out. The theoretical linear case and the practical MLP case are two instantiations of the same function-approximation framework that underlies our algorithm.
> >
> > - **Theoretical side (linear model).**
> >
> >   The linear approximation is a standard and analytically tractable setting that allows us to formally establish that our algorithm performs well under function approximation. This setting enables us to derive finite-sample guarantees and provide theoretical understanding of the selection process.
> >
> > - **Practical side (MLP model).**
> >
> >   The MLP serves as a more expressive approximation class for modeling the relationship between embeddings and objectives in real applications. While more flexible, it fits naturally into the same framework: the algorithm only requires a function approximator that predicts objective values from embeddings.
> >
> > Overall, these two cases illustrate complementary aspects of our approach: the linear model provides theoretical grounding, while the MLP demonstrates practical applicability within the same overarching function-approximation view.
> >
> > ---
> >
> > **Q5** Missing discussion on scalability with number of objectives $m > 2$
> >
> > **A5** We have included the following discussion in the revision in Appendix B.2 of the revision.
> >
> > Our framework extends naturally to settings with more than two objectives. The key quantities underlying our algorithms, such as the Pareto gaps and constrained gaps, can be generalized to $m$-dimensional objective spaces without changing the fundamental structure of the elimination or selection rules. The computational overhead associated with these extensions is modest: dominance checks and gap computations scale polynomially with $m$ and remain efficient for the small number of objectives typically encountered in practice. Thus, the proposed algorithms retain both conceptual simplicity and computational tractability when applied to $m > 2$ multi-objective prompt selection problems.
> >
> > ---
> >
> > **Q6** Prompt generation via LLM + manual filtering introduces selection bias
> >
> > **A6** We thank the reviewer for this comment. We would like to clarify that our methodology focuses on *prompt selection* from a fixed candidate pool, and is therefore **agnostic to how that pool is generated**. Any bias introduced during prompt generation *does not affect the validity of our algorithms or theoretical guarantees*, which make no assumptions about the pool's origin or quality.
> >
> > In our experiments, we use a simple LLM-based generation plus light filtering as a reasonable and practical choice to obtain a diverse prompt set, but *this step is not intrinsic to our method*. In practice, the resulting pool still spans a broad range of qualities across the objectives, ensuring that the selection task remains meaningful.

---

> > > ### Author Response · Authors · 2025-11-21
> > >
> > > **Q7** What specific advantages does the bandit formulation provide over off-the-shelf HPO methods like Bayesian optimization for this problem?
> > >
> > > **A7** We thank the reviewer for this question. The bandit formulation provides several advantages over off-the-shelf HPO methods such as Bayesian optimization (BO) in our setting:
> > >
> > > - **Direct alignment with the problem structure.**
> > >   Our setting involves repeated, noisy evaluations of prompts under a strict evaluation budget. Bandit algorithms are specifically designed for this **sequential, stochastic feedback** model, whereas standard BO assumes smoother objective landscapes and typically relies on richer function structure than we can exploit with LLM-based evaluations.
> > >
> > > - **Multi-objective + constrained setting.**
> > >   Off-the-shelf BO methods do not directly address multi-objective or constrained problems in a way that scales to noisy, discrete candidate pools. Substantial modifications would be required, and to our knowledge there is no established BO baseline for multi-objective prompt optimization. In contrast, our bandit-style formulation naturally accommodates these requirements and comes with theoretical guarantees.
> > >
> > > - **Practical simplicity and robustness.**
> > >   Elimination-based bandit algorithms require minimal tuning and no assumptions about kernels, priors, or smoothness. BO, by contrast, typically requires selecting kernel structures, setting priors, and performing expensive posterior updates or sampling steps, which may become brittle when reward evaluations come from stochastic LLM outputs. In practice, this makes bandit methods much easier to deploy reliably.
> > >
> > > In summary, the bandit formulation directly matches the stochastic, budget-limited nature of the prompt-selection problem and avoids the modeling assumptions and complexity required by traditional BO methods.
> > >
> > > ---
> > >
> > > **Q8** Can you demonstrate the method on real safety or fairness constraints?
> > >
> > > **A8** We thank the reviewer for the suggestion. Currently we are not aware of any publicly available prompt sets or evaluation benchmarks that provide prompt-level, quantitative, and reliable safety or fairness objectives suitable for multi-objective prompt selection. Common safety datasets (e.g., RLHF helpfulness/harmlessness data) contain response-level preference labels rather than a candidate instruction-prompt pool or per-prompt safety metrics, and adapting them would require constructing new prompts and training a reward model from preferences, which is essentially a full RLHF pipeline outside our scope. Likewise, fairness benchmarks such as BBQ report dataset-level bias/accuracy metrics that require evaluating thousands of examples per prompt and do not provide an instruction-prompt pool, making them incompatible with our strict evaluation budget and fixed-pool setting. Demonstrating our method under real safety or fairness objectives would therefore require building new datasets, metrics, and evaluation pipelines. We view this as an important direction for future work.
> > >
> > > -------------
> > >
> > > We thank the reviewer again for the helpful comments and suggestions for our work. If our response resolves your concerns to a satisfactory level, we kindly request the reviewer to consider raising the rating of our work. Certainly, we are more than happy to address any further questions that you may have.

---

> > > > ### Author Response · Authors · 2025-11-27
> > > > **A gentle reminder**
> > > >
> > > > Dear Reviewer,
> > > >
> > > > We've taken your initial feedback into careful consideration in our responses. Could you please check whether our responses have properly addressed your concerns? If so, we would greatly appreciate your consideration in increasing your initial score. Certainly, we are more than happy to answer any further questions.
> > > >
> > > > Thank you for your time and effort in reviewing our work!
> > > >
> > > > Best Regards,
> > > >
> > > > Authors

---

### Official Review · Reviewer_BAtQ · 2025-10-30

**Soundness:** 3
**Presentation:** 2
**Contribution:** 2
**Rating:** 6
**Confidence:** 2

**Summary:**

The paper frames multi-criteria prompt selection as pure-exploration multi-objective bandits, targeting both best feasible prompt and Pareto set recovery. It proposes two elimination frameworks GENSEC and GENPSI respectively. In the linear case, the paper proves an exponentially decaying misidentification bound with only logarithmic dependency in K. Experiments on XSum and CNN/DailyMail with Llama-3-8B-Instruct and Gemma-7B-IT show consistent gains over a uniform baseline.

**Strengths:**

The paper moves beyond single-metric prompt selection to an explicit multi-objective setup, which is well motivated for practical problem.

The paper provide theoretical error bound for linear rewards.

**Weaknesses:**

In experiments, comparisons are largely to a uniform allocator,  there is no multi-objective prompt optimization baselines, such as weighted sums with single-objective BAI or recent evolutionary multi-objective prompt optimizers mentioned in sec. 2.

HV is “normalized by the ground-truth Pareto set,” but the procedure to obtain that ground truth is not fully specified.

Typo: in line 128, should be  "Let $x$ be a prompt"

**Questions:**

How exactly is the “ground-truth Pareto set” computed for normalization?

Can you add weighted sum +BAI, and evolutionary multi-objective prompt optimizer as baselines?

---

> ### Author Response · Authors · 2025-11-21
>
> We appreciate the reviewer’s thoughtful comments. Below, we summarize the questions and provide our responses. We have updated the submission accordingly and highlighted the major changes in blue.
>
> ---
>
> **Q1** In experiments, comparisons are largely to a uniform allocator; there are no multi-objective prompt optimization baselines, such as weighted sums with single-objective BAI or recent evolutionary multi-objective prompt optimizers mentioned in sec. 2.
>
> **A1** We thank the reviewer for raising this point. Below we clarify why the two mentioned baselines were not included in our main comparisons.
>
> - **Weighted sums with single-objective BAI.**
>   This approach first collapses the multi-objective problem into a scalar objective and then returns a single best prompt for a chosen weight vector. To approximate a Pareto set or achieve a competitive hypervolume, one would need to run many BAI instances with different weights, which either (i) multiplies the total number of LLM calls or (ii) splits the fixed evaluation budget across weights, substantially degrading performance. In the constrained setting, feasibility is also not guaranteed, since it depends on the specific weight choice. For these reasons, we did not treat weighted-sum BAI as a directly comparable baseline.
>
> - **Evolutionary multi-objective prompt optimizers.**
>   As noted in Sec. 2, evolutionary optimizers operate in a different regime: they generate and mutate prompts across iterations, whereas our methods assume a fixed candidate prompt pool and focus solely on efficient multi-objective selection within that pool. A direct comparison would therefore conflate the quality of the generation mechanism with the quality of the selection algorithm and depend heavily on the specific generator used. We view evolutionary methods as complementary rather than competing. A promising future direction is to combine them, e.g., using our algorithms as the selection module within each evolutionary step.
>
> ---
>
>
> **Q2** HV is “normalized by the ground-truth Pareto set,” but the procedure to obtain that ground truth is not fully specified. How exactly is the “ground-truth Pareto set” computed for normalization?
>
> **A2** We thank the reviewer for pointing out the ambiguity. By “normalized by the ground-truth Pareto set,” we mean that the reported HV is computed as the ratio between (i) the hypervolume of the set recovered by an algorithm and (ii) the hypervolume of the true Pareto set obtained by exhaustively evaluating all prompts in the candidate pool. Since the entire pool is finite, we can directly compute its exact Pareto front and hypervolume. We {have clarified} this definition in the revision.
>
> ---
>
>
> **Q3** Can you add weighted sum +BAI, and evolutionary multi-objective prompt optimizer as baselines?
>
> **A3** We thank the reviewer for the suggestion. As discussed in **A1**, we did not include weighted-sum + BAI or evolutionary multi-objective prompt optimizers because they are not directly comparable to our setting and introduce additional assumptions and design choices (e.g., scalarization weights or prompt-generation mechanisms). Our methods assume a fixed candidate prompt pool and a fixed evaluation budget, whereas these baselines operate under different regimes. For this reason, we did not treat them as main baselines in this version.
>
> If the reviewer has specific variants they believe can be meaningfully adapted to our setting, we would be happy to consider adding them in the revision.
>
> ---
>
>
> **Q4** Typo: in line 128, should be ”Let $x$ be a prompt”
>
>
> **A4** We thank the reviewer for catching this typo. We have corrected it in the revision.
>
> ------
>
> We thank the reviewer again for the helpful comments and suggestions for our work. We hope that our response resolves your concerns to a satisfactory level, and we are more than happy to address any further questions that you may have.

---

### Official Review · Reviewer_NCGw · 2025-11-03

**Soundness:** 3
**Presentation:** 2
**Contribution:** 2
**Rating:** 4
**Confidence:** 3

**Summary:**

This paper investigates the multi-objective prompt selection problem under two settings: Pareto prompt set recovery and best feasible prompt identification. The authors treat the prompt selection problem as a bandit problem and develop a multi-objective bandit algorithm. The experimental results demonstrate that the proposed method outperforms baseline methods on two tasks.

**Strengths:**

- Focusing on multiple evaluation criteria in prompt selection is convincing, as it reflects practical scenarios better than single-objective settings.
- The experimental evaluation demonstrates that the proposed method outperforms baseline methods on two tasks.

**Weaknesses:**

- The proposed method requires a pre-defined candidate prompt set, and its performance is limited by the quality of the candidate prompt set. However, how to construct a good candidate prompt set is not discussed in this paper.
- In the experiment, the candidate prompt pool is up to 100, which seems relatively small. It is unclear whether the proposed method works well when the candidate prompt pool is large.
- The budget $B$ considered in the experiment is small (up to 10). How is the performance of the proposed method when the budget and candidate prompt pool size are large?
- The experimental comparison is conducted only with a simple baseline and the variants of the proposed method. Comparison with existing prompt optimization methods may be helpful in clarifying the absolute effectiveness of the proposed method.
- Only two datasets (XSum and CNN/DailyMail) are considered in the experiment. It would be better to include more datasets to verify the effectiveness of the proposed method.

**Questions:**

- How is the performance of the proposed method when the budget and candidate prompt pool size are large?
- Could you explain the reason that the current experimental settings are chosen (e.g., datasets and baselines)?
- Please describe a practical use case where the proposed method is more effective than other existing prompt optimization (generation) methods.

---

> ### Author Response · Authors · 2025-11-21
>
> We appreciate the reviewer’s thoughtful comments. Below, we summarize the questions and provide our responses. We have updated the submission accordingly and highlighted the major changes in blue.
>
> ---
>
> **Q1** The proposed method requires a pre-defined candidate prompt set, and its performance is limited by the quality of the candidate prompt set. However, how to construct a good candidate prompt set is not discussed in this paper.
>
> **A1** We appreciate the reviewer’s comment and clarify the role of the candidate prompt set in our framework.
>
> First, our work focuses specifically on the *selection* problem given a finite candidate prompt set. Our goal is not to design or optimize the prompt-generation process itself, but to study how to reliably select high-quality prompts from any such set, regardless of whether it is strong or weak.
>
> Second, from a theoretical perspective, all of our algorithms and guarantees are fully **agnostic to how the candidate set is obtained**. They apply to any finite prompt pool, independent of its construction.
>
> Third, we would like to emphasize that our method is compatible with the prompt engineering pipeline in realistic scenarios:
>
> (1) *As a selection layer over evolutionary prompt-generation methods:* Many evolutionary algorithms begin with a noisy or low-quality initial pool and iteratively refine it. At each iteration, our method can select a high-quality Pareto or feasible subset under multiple objectives, providing a principled selection mechanism within such pipelines.
>
> (2) *As a selector for strong, fixed prompt pools:* In other settings, a high-quality candidate pool may already be available from prior optimization runs or strong baselines. When adapting such a pool to a new user, domain, or constraint set, our algorithm can be directly applied to identify a tailored Pareto or feasible subset without the need to regenerate prompts.
>
> Finally, in our experiments, we instantiate the construction of the candidate set by first prompting an LLM with a few illustrative examples to generate candidate prompts, and then manually filtering out a small number of clearly irrelevant or incorrect prompts. The prompt generation process is described at the beginning of the experiments section.
>
> ---
>
> **Q2** In the experiment, the candidate prompt pool is up to 100, which seems relatively small. It is unclear whether the proposed method works well when the candidate prompt pool is large.
> How is the performance of the proposed method when the candidate prompt pool sizes are large?
>
> **A2** Thank you for raising this point. We have performed further experiments using a significantly larger prompt pool (1,000 prompts) under 5 random seeds. The results are reported in Figure 8 in Appendix C.5 of the revision, which show that our methods remain consistently stronger than the uniform-pulling baseline.
>
> ---
>
> **Q3** Considered in the experiment is small (up to 10). How is the performance of the proposed method when the budgets are large?
>
> **A3** We have performed additional experiments for the Llama-3 + XSum model–dataset pair under budgets of 20 and 30 evaluations per arm, and reported the results in Figure 7 in Appendix C.5 of the revision. As expected, when the evaluation budget increases, all methods, including the baseline, get closer to the optimal performance. Nevertheless, our algorithms consistently remain above the baseline.

---

> > ### Author Response · Authors · 2025-11-21
> >
> > **Q4** The experimental comparison is conducted only with a simple baseline and the variants of the proposed method. Comparison with existing prompt optimization methods may be helpful in clarifying the absolute effectiveness of the proposed method.
> >
> > **A4** We thank the reviewer for noting that a richer set of baselines could further strengthen our empirical evaluation. We agree with this general direction and clarify below why we selected our current baseline and why certain alternative methods were not included.
> >
> > First, we chose the uniform allocator as our main baseline because it operates under *exactly* the same assumptions as our methods: a fixed candidate prompt pool and a fixed evaluation budget, with no additional modeling choices. This makes it a clean, controlled, and directly comparable baseline for isolating the effect of our selection algorithms.
> >
> > Second, we examined two representative classes of existing prompt-optimization methods but found fundamental mismatches with our setting:
> >
> > - *Weighted-sum + single-objective BAI:*
> >   This approach collapses the vector reward into a scalar and returns a single best prompt for each weight. To approximate a Pareto set or compute a meaningful hypervolume, one would need to run multiple BAI instances with different weights, which either multiplies the total number of LLM calls or drastically reduces the budget per weight. Moreover, this approach provides no principled mechanism for enforcing constraints, which is essential in our formulation.
> >
> > - *Evolutionary multi-objective prompt optimizers:* These methods repeatedly generate and mutate prompts, while our problem assumes a fixed finite candidate pool and focuses on efficient selection. Comparing with evolutionary methods would conflate generation quality with selection performance and depend heavily on the specific generator used, making the comparison neither clean nor directly aligned with our setting.
> >
> > Given these considerations, we were unable to identify existing methods that are directly comparable under the same assumptions. That said, we are fully open to expanding our empirical study: if there are relevant baselines we may have overlooked, we would greatly appreciate the reviewer pointing them out, and we will be happy to include them in the revision.
> >
> > ---
> >
> > **Q5** Could you explain the reason that the current experimental settings are chosen (e.g., datasets and baselines)?
> >
> > **A5** We thank the reviewer for the question. Our choices of datasets and baselines are motivated by the following considerations:
> >
> > **Datasets \& Objectives**
> >
> > - *Summarization as a standard testbed.*
> >   We use summarization because it is a well-established benchmark for prompt optimization, with reliable datasets and automatic metrics that allow controlled large-scale experiments.
> >
> > - *ROUGE-L vs. brevity as a meaningful conflicting objective pair.*
> >   These two objectives capture a natural informativeness–conciseness trade-off and produce non-trivial Pareto fronts or constraint boundaries. This makes them well-suited for evaluating multi-objective selection algorithms.
> >
> > - *Challenges in constructing comparable multi-objective benchmarks for other tasks.*
> >   For tasks involving safety, factuality, or latency, multi-objective evaluation typically requires task-specific metrics, expert annotation, or large-scale human evaluation. These are difficult to reproduce consistently at scale and are beyond our current experimental capacity. For this reason, we begin with summarization, where high-quality automatic metrics are available.
> >
> > **Baselines**
> >
> > - *Baselines matched to our setting.*
> >   Our methods assume a fixed candidate prompt pool and a fixed evaluation budget. We therefore select baselines that operate under exactly the same conditions. The uniform allocator is the simplest such baseline and serves as a clean control that isolates the gain from active selection.
> >
> > - *Why certain existing prompt-optimization methods were not used as main baselines.*
> >   As elaborated in our response **A4**,
> >   *weighted-sum + single-objective BAI* collapses multi-objective rewards into a scalar and produces only a single prompt per run. Approximating a Pareto front requires many such runs and does not handle constraints well.
> >   *Evolutionary multi-objective prompt optimizers* repeatedly generate or mutate prompts, which differs fundamentally from our setting of selection from a fixed finite pool. A direct comparison would conflate generation quality with selection performance.
> >
> > We hope these clarifications help explain the rationale behind our experimental design.

---

> > > ### Author Response · Authors · 2025-11-21
> > >
> > > **Q6** Only two datasets (XSum and CNN/DailyMail) are considered in the experiment. It would be better to include more datasets to verify the effectiveness of the proposed method.
> > >
> > > **A6** We thank the reviewer for the suggestion. We use XSum and CNN/DailyMail because they are standard summarization benchmarks with reliable metrics, and our method performs well on both. Due to time and computational constraints, we cannot add more large-scale datasets at this point, but we would like to clarify that our approach is **dataset-agnostic** and can be naturally extended to additional benchmarks.
> > >
> > > ---
> > >
> > > **Q7** Please describe a practical use case where the proposed method is more effective than other existing prompt optimization (generation) methods.
> > >
> > > **A7** We thank the reviewer for the comment. We highlight two practical scenarios where our method is particularly suitable:
> > >
> > > - **Re-using an existing prompt pool.**
> > >   When a large candidate prompt set has already been generated offline by another optimizer, re-running that optimizer in a new environment can be very costly. Our method can operate directly on this fixed pool to efficiently identify a new Pareto or feasible-optimal subset under the new objectives with a much smaller evaluation budget.
> > >
> > > - **Selection module in evolutionary frameworks.**
> > >   Evolutionary prompt optimizers repeatedly select “survivor’’ prompts across generations under multiple objectives. Our algorithm can serve as a principled multi-objective selection component for this step, improving sample efficiency compared to heuristic selection rules when evaluations are expensive.
> > >
> > > --------
> > >
> > > We thank the reviewer again for the helpful comments and suggestions for our work. If our response resolves your concerns to a satisfactory level, we kindly request the reviewer to consider raising the rating of our work. Certainly, we are more than happy to address any further questions that you may have.

---

> > > > ### Author Response · Authors · 2025-11-27
> > > > **A gentle reminder**
> > > >
> > > > Dear Reviewer,
> > > >
> > > > We've taken your initial feedback into careful consideration in our responses. Could you please check whether our responses have properly addressed your concerns? If so, we would greatly appreciate your consideration in increasing your initial score. Certainly, we are more than happy to answer any further questions.
> > > >
> > > > Thank you for your time and effort in reviewing our work!
> > > >
> > > > Best Regards,
> > > >
> > > > Authors

---

### Author Response · Authors · 2025-12-03

We thank the AC for handling our submission.  Following the recent ICLR policy change, for the convenience of the new AC, we first summarize our main contributions below.

- **Novel problem formulation and unified framework.**
  As the reviewers noted, the paper introduces a new problem setting: multi-objective prompt selection with both Pareto-style and hard-constraint objectives, and provides a unified bandit-based framework for addressing it. To our knowledge, this is the first application of bandit algorithms to multi-objective prompt optimization, and also the first to incorporate explicit hard constraints. These problem formulations are themselves new and require adapting and integrating concepts that do not appear in standard prompt optimization or multi-armed bandit literature.

- **New theoretical results under linear function approximation.**
  The theoretical analysis of GENSEC under linear function approximation is not available in prior work. Establishing guarantees in this setting is non-trivial. Due to the multi-objective setting, three types of arms need to be analyzed carefully: feasible but suboptimal arms, deceiver arms (infeasible but better than the optimal feasible arm on the primary objective), and infeasible sub-optimal arms. Handling these cases introduces substantially more complex failure modes than in prior single-objective analyses and is a key technical novelty of our proof.
  The theoretical result also provides conceptual support for the non-linear (MLP-based) practical instantiation, since both models share the same function approximation perspective.

- **Demonstrated empirical effectiveness.**
  Across experiments, including the newly added results with 1,000 prompts in the revision, our algorithms consistently outperform the uniform baseline and achieve strong approximations of the Pareto sets under tight evaluation budgets.
  These results suggest that the proposed framework is not only theoretically grounded but also **practically effective** in realistic prompt-selection scenarios.

Next, we summarize the major concerns of the reviewers, and highlight how we have addressed those concerns during the rebuttal.

- **Scale of the experiments.** Reviewers NCGw, pHhY, i6FQ raised concerns regarding the size of the candidate prompt pool ($<100$) in our original submission being relatively small. We have performed further experiments using a significantly larger prompt pool (1,000 prompts) under 5 random seeds. The results are reported in Figure 8 in Appendix C.5 of the revision, which show that our methods remain consistently stronger than the uniform-pulling baseline. Besides, we have also performed additional experiments with higher budgets ($B=20,30$) and varying constraints, and reported the results in Figure 7 and Figure 9 in the revision.

- **Datasets and objectives.** Reviewers NCGw and i6FQ suggested extending the experiments to tasks beyond summarization and to additional objectives. As we explained in the response, there is currently a lack of suitable multi-objective benchmarks outside summarization; constructing such benchmarks would require substantial task design, metric development, and large-scale human or model evaluations, which is beyond the scope of this work.
  At the same time, summarization is one of the most widely used benchmarks in prompt-optimization research, and the objective pair (ROUGE, brevity) provides a clean and non-trivial trade-off between informativeness and conciseness, making it a natural testbed for evaluating multi-objective prompt selection.

  Nevertheless, our framework itself is task- and metric-agnostic and can be applied to a broad range of tasks with diverse measurable objectives, such as safety, factuality, latency, or monetary cost.

- **Limited baselines for comparison.** Reviewers NCGw, BAtQ, and pHhY suggested including additional prompt-optimization baselines. To our knowledge, no existing methods operate under the same setting as ours. For instance, evolutionary multi-objective prompt-search approaches are not directly comparable, as they repeatedly generate or mutate prompts rather than selecting from a fixed pool.
  Nevertheless, our algorithms can serve as a principled and sample-efficient selection module within each evolutionary step.

- **Connection to hyperparameter optimization (HPO).** Reviewer pHhY noted that the prompt-selection problem is essentially a gradient-free HPO problem. We clarified that prompt selection differs from standard HPO in that it optimizes over a discrete set rather than a continuous parameter space, and that LLM-based objective evaluations are typically stochastic. Consequently, a bandit-based approach is a more suitable solution.

---

> ### Author Response · Authors · 2025-12-03
>
> - **Definition of hypervolume gain.** Reviewers BAtQ and i6FQ expressed confusion about the definition of hypervolume (HV) gain. We clarified the definition and highlighted that, under our algorithms, the percentage of HV recovered relative to the true Pareto front is consistently higher than that of the uniform baseline.
>
> We respectfully request that the AC takes these updates and responses into account in the final evaluation. Thank you for your support.
>
> Best regards,
> Authors

---

### Meta-Review · Area_Chair_KzGE · 2025-12-28

**Summary:**

The reviewers generally acknowledged the novelty of applying bandit algorithms to multi-objective prompt selection with hard constraints. However, significant concerns were raised regarding the experimental design. Specifically, reviewers NCGw, pHhY, and i6FQ questioned the small scale of the initial experiments (prompt pool size and budget). Multiple reviewers (NCGw, BAtQ, pHhY) criticized the lack of comparison against stronger baselines, such as evolutionary methods or weighted-sum BAI. Reviewers NCGw and i6FQ also expressed concern over the narrow scope of the evaluation, which was limited to summarization tasks and two metrics (ROUGE and brevity), suggesting the inclusion of safety or fairness objectives. Reviewer pHhY questioned the contribution's distinction from standard Hyperparameter Optimization (HPO), and Reviewer i6FQ questioned the algorithmic novelty, viewing the method as a wrapper around existing bandit algorithms.

**Reviewer Concerns:**

**Concerns Addressed:**
* **Experimental Scale:** The concern regarding the small size of the candidate prompt pool and evaluation budget (raised by **NCGw**, **pHhY**, **i6FQ**) was effectively addressed. The authors conducted new experiments with 1,000 prompts and increased budgets, demonstrating that their method continues to outperform the uniform baseline.
* **Hypervolume Definition:** The confusion regarding the "ground-truth Pareto set" normalization (raised by **BAtQ**, **i6FQ**) was clarified by the authors.
* **Theory vs. Practice:** The gap between the linear theoretical analysis and the MLP practical implementation (raised by **pHhY**) was explained as complementary instantiations of the same function-approximation framework.

**Concerns Outstanding:**
* **Baselines:** The request for comparisons against other prompt optimization methods like evolutionary algorithms or InstructZero (raised by **NCGw**, **BAtQ**, **pHhY**) remains largely outstanding. The authors argued these methods operate under different settings (generation vs. selection from a fixed pool), but did not provide the adapted comparisons requested by reviewers to establish absolute effectiveness.
* **Task/Objective Diversity:** The request to expand beyond summarization to tasks like safety or fairness (raised by **NCGw**, **i6FQ**) was not empirically addressed. The authors justified this based on the lack of suitable benchmarks and high computational costs, leaving the empirical scope limited to one domain.
* **Ablations:** While promised, the sensitivity analysis on elimination margins and thresholds requested by **i6FQ** was not fully detailed in the provided text.

**Reviewer Scores:**

* **Reviewer NCGw (4 -> 6):** This reviewer's primary concerns were the small prompt pool and budget. The authors directly addressed these "Questions" by providing new results with 1,000 prompts and larger budgets. These additional experiments would likely increase the score.
* **Reviewer BAtQ (6 -> 6):** This reviewer was already leaning positive. While they requested additional baselines which were not added, the authors provided a clear rationale for the omission and clarified the hypervolume definition. The score likely remains unchanged.
* **Reviewer pHhY (2 -> 4):** This reviewer had strong objections regarding the contribution relative to HPO and the experimental scale. The authors directly satisfied the request for "experiments with 1000+ initial prompts." However, the fundamental disagreement about the novelty compared to HPO and the lack of generation-based baselines likely prevents a full positive shift, moving the score to borderline.
* **Reviewer i6FQ (6 -> 6):** This reviewer found the scope narrow and the novelty limited. While the authors defended their position and clarified interpretation of the results, they did not expand the task scope as requested. The score likely remains stable as the core critique of "narrow scope" persists.

---

### Decision · Program_Chairs · 2026-01-26

Accept (Poster)